# End-to-End Mesh Optimization of a Hybrid Deep Learning Black-Box PDE Solver

## Abstract

Deep learning has been widely applied to solve partial differential equations (PDEs) in computational fluid dynamics. Recent research proposed a PDE correction framework that leverages deep learning to correct the solution obtained by a PDE solver on a coarse mesh. However, end-to-end training of such a PDE correction model over both solver-dependent parameters such as mesh parameters and neural network parameters requires the PDE solver to support automatic differentiation through the iterative numerical process. Such a feature is not readily available in many existing solvers. In this study, we explore the feasibility of end-to-end training of a hybrid model with a black-box PDE solver and a deep learning model for fluid flow prediction. Specifically, we investigate a hybrid model that integrates a black-box PDE solver into a differentiable deep graph neural network. To train this model, we use a zeroth-order gradient estimator to differentiate the PDE solver via forward propagation. Although experiments show that the proposed approach based on zeroth-order gradient estimation underperforms the baseline that computes exact derivatives using automatic differentiation, our proposed method outperforms the baseline trained with a frozen input mesh to the solver. Moreover, with a simple warm-start on the neural network parameters, we show that models trained by these zeroth-order algorithms achieve an accelerated convergence and improved generalization performance.

## 1 Introduction

Studying the fluid dynamics behavior is a fundamental and challenging pursuit in the physical sciences due to the complex nature of solving highly nonlinear and large-scale Navier-Stokes partial differential equations (PDEs) Batchelor (1967). Conventionally, numerical solvers employ finite-difference methods to approximate PDE solutions on discrete meshes. However, this approach requires running a full simulation for each mesh, which can impose a substantial computational burden, particularly when dealing with highly granular meshes.

To address these challenges, there has been a growing interest in the application of data-driven deep learning techniques for inferring solutions to PDEs, particularly within the field of computational fluid dynamics (CFD) (Afshar et al., 2019; Guo et al., 2016; Wiewel et al., 2018; Um et al., 2017; Belbute-Peres et al., 2020). Notably, one line of research aims to predict the outcomes of physical processes directly using data-driven deep learning methods (Afshar et al., 2019; Guo et al., 2016). However, these approaches do not consider the underlying physical mechanisms and often require a substantial volume of data, particularly high-granularity simulation data. Consequently, the learned models often exhibit limited generalizability when solving PDEs with unseen parameters.

Recently, one promising approach has emerged to further enhance the generalization performance (Belbute-Peres et al., 2020). This approach proposes to integrate a physical PDE solver into a graph neural network, wherein the solver generates simulation data at a coarse granularity to aid the network in predicting simulation outcomes at a fine granularity. Notably, the solver is required to be differentiable, which enables the simultaneous optimization of both the coarse mesh and network parameters using stochastic gradient optimizers. Despite demonstrating significant enhancement in generalization performance, this approach requires that the solver supports automatic differentiation – an attribute lacking in many existing solvers.

This limitation arises from the fact that some solvers are tailored to specific application domains and are complied by outdated languages, demanding substantial effort and interdisciplinary expertise to rewrite them with automatic differentiation capability. On the other hand, since the optimization of the coarse mesh is considerably less complex than optimizing the neural network parameters, it is possible to conduct mesh optimization using noisy algorithms that do not rely on exact gradient queries. Inspired by these insights, we are motivated to study the following key question.

- *Q: Can we perform end-to-end deep learning for fluid flow prediction with black-box solvers that do not support automatic differentiation?*

In this work, we provide an affirmative answer to the above question by developing a zero-order type algorithm that can optimize the deep model's parameters and solver's mesh parameters end-to-end without back-propagation through the solver. We summarize our contributions as follows.

## 1.1 Our Contributions

We consider the CFD-GCN hybrid machine learning model proposed in Belbute-Peres et al. (2020) for fluid flow prediction. In particular, this hybrid model involves a differentiable PDE solver named SU2 (Economon et al., 2015). However, in many practical situations, exposing new variables for differentiation in an external module requires either highly complicated modification on the underlying codes or is simply not supported for closed-sourced commercial software. To overcome this obstacle, our goal is to train this hybrid model without querying differentiation through the PDE solver.

1. Based on the differentiable CFD-GCN hybrid model, we develop various algorithms that can train this model without directly querying the exact gradients of the solver. Consequently, one can integrate any black-box solvers into the hybrid system and apply our algorithms to train the system parameters.

2. Specifically, we apply two classical zeroth-order gradient estimators, i.e., Coordinate-ZO and Gaussian-ZO (see eqs. (5) and (6)) to estimate the gradients of the solver solely via forward-propagation, and use them together with the neural network's gradients to train the hybrid model. Our experiments on fluid flow prediction verify the improved generalization performance of these zeroth-order algorithms compared with a first-order gradient baseline trained using a frozen input mesh to the solver. Furthermore, we propose a mixed gradient estimator named Gaussian-Coordinate-ZO (see eq. (7)) that combines the advantages of the previous two zeroth-order estimators, and show that it leads to an improved generalization performance.

3. Lastly, we observe an asymmetry between the optimization of mesh parameters and that of neural network model parameters, based on which we propose a warm-start strategy to enhance the training process. Experiments show that such a simple strategy leads to an improved generalization performance.

## 1.2 Related Work

**Deep learning for CFD.** The incorporation of machine learning into Computational Fluid Dynamics (CFD) has been a topic of growing interest in recent years. In many cases, the deep learning techniques are directly used to predict physical processes Afshar et al. (2019); Guo et al. (2016). Some recent studies combine the physical information and deep learning model to obtain a hybrid model Belbute-Peres et al. (2020); Um et al. (2020). This paper goes one step further to consider the scenario that the external physical features extractor is black-box.

**Differentiability of Solvers.** Fluid dynamics simulators, like SU2 (Economon et al., 2015), have incorporated adjoint-based differentiation and are extensively used in fields such as shape optimization (Jameson, 1988) and graphics (McNamara et al., 2004). However, in physics and chemistry disciplines, ML models may be required to interface with numerical solvers or complex simulation codes for which the underlying

systems are non-differentiable Thelen et al. (2022); Tsaknakis et al. (2022); Louppe et al. (2019); Abreu de Souza et al. (2023); Baydin et al. (2020).

**Zeroth-order optimization.** The adoption of zeroth-order optimization techniques has garnered considerable attention within the domain of optimization research. Instead of directly evaluating the function derivative, the zeroth-order method applies the random perturbation to the function input to obtain the gradient estimation (Liu et al., 2018b; Duchi et al., 2015; Liu et al., 2020). Based on the types of random direction, the zeroth-order estimator can be roughly classified into Gaussian estimator (Berahas et al., 2022; Nesterov & Spokoiny, 2017) and coordinate estimator (Berahas et al., 2022; Kiefer & Wolfowitz, 1952). This paper mainly focuses on the performance of these classical zeroth-order optimization techniques in the hybrid model.

**Deep operator learning.** Recent research has made significant strides in applying deep learning methods to solve partial differential equations (PDEs), aiming to learn mappings between infinite-dimensional function spaces. The development of Deep Operator Networks (DeepONet) (Lu et al., 2021) leverages the universal approximation theorem to model complex nonlinear operators and Mionet (Jin et al., 2022) extends it to multiple input functions. The Multigrid-augmented deep learning preconditioners for the Helmholtz equation represent another innovative approach, integrating convolutional neural networks (CNNs) with multigrid methods to achieve enhanced efficiency and generalization (Azulay & Treister, 2022). Furthermore, the Fourier Neural Operator (FNO) (Li et al., 2020) offers another method by parameterizing the integral kernel directly in Fourier space, thereby achieving improved performance on benchmarks like Burgers' equation and Navier-Stokes equation with considerable speed advantages over traditional solvers and Geo-FNO (Li et al., 2022) generalizes this method to general meshes. Additionally, the introduction of the GNOT (Ying et al., 2023) and CFDNet (Obiols-Sales et al., 2020) brings forward a hybrid approach that combines deep learning with traditional numerical methods, enhancing the computational efficiency and accuracy of PDE solutions. Compared to these hybrid methods, we are particularly interested in the CFD-GCN model (Belbute-Peres et al., 2020) and its variant since they additionally require the optimization step on the parameters in the external solvers, which are often non-differentiable. We aim to design an alternative differentiation method to replace the complex implementation of auto-differentiation of external PDE solvers.

## 2 Background: Hybrid Machine Learning for Fluid Flow Prediction

In this section, we recap the hybrid machine learning system named CFD-GCN proposed by Belbute-Peres et al. (2020) for fluid flow prediction. We first introduce the SU2 PDE solver developed for computation fluid dynamics. Then, we introduce the CFD-GCN model that integrates this PDE solver into a graph convolutional neural network. Lastly, we discuss the differentiability issue of hybrid machine learning systems.

### 2.1 The SU2 PDE Solver

SU2 is a numerical solver developed for solving PDEs that arise in computation fluid dynamics Economon et al. (2016). SU2 applies the finite volume method (FVM) to solve PDEs and calculate certain physical quantities over a given discrete mesh. In particular, we consider SU2 because it supports automatic differentiation, and hence provides a first-order gradient baseline to compare with our zeroth-order algorithms.

To explain how SU2 works, consider the task of calculating the airflow fields around an airfoil. The input to SU2 includes the following three components.

1. *Mesh*: a discrete approximation of the continuous flow field. The mesh consists of multiple nodes whose positions depend on the shape of the airfoil;

2. *Angle of attack (AoA)*: a parameter that specifies the angle between the chord line of the airfoil and the oncoming airflow direction;

3. *Mach number*: a parameter that specifies the ratio of the airfoil's speed to the speed of sound in the same medium.

Given the above input and initial conditions, SU2 can calculate the air pressure and velocity values at each node of the mesh by solving the following Navier-Stokes PDEs:

$$\frac{\partial V}{\partial t} + \nabla \cdot \bar{F}^c(V) - \nabla \cdot \bar{F}^v(V, \nabla V) - S = 0$$

where AoA is used to define the boundary condition (the flow-tangency Euler wall boundary condition for airfoil and the standard characteristic-based boundary condition for the farfield) and Mach number is used to define the initial condition to describe the initial velocity. However, one issue of using a numerical solver is that the runtime scales polynomially with regard to the number of nodes in the mesh. For example, for a standard airfoil problem, the run time of SU2 is about two seconds for a coarse input mesh with 354 nodes, and it increases to more than three minutes for a fine input mesh with 6648 nodes (Belbute-Peres et al., 2020). Therefore, it is generally computationally costly to obtain a simulation result at a high resolution, which is much desired due to its better approximation of the original PDEs over the continuous field. Another issue is the lack of generalizability, i.e., one needs to run SU2 to obtain the simulation result for each instance of the AoA and mach number parameters.

## 2.2 The CFD-GCN Learning System

To accelerate SU2 simulations, Belbute-Peres et al. (2020) developed CFD-GCN – a hybrid machine learning model that integrates the physical SU2 solver into a graph convolution network (GCN). This hybrid model aims to predict the SU2 simulation outcome associated with fine mesh using that associated with coarse mesh, as illustrated in Figure 1. In particular, since the coarse mesh usually contains very few nodes, it is critical to jointly optimize the coarse mesh's node positions with the GCN model parameters. For more details, please refer to the Figure 1 in Belbute-Peres et al. (2020).

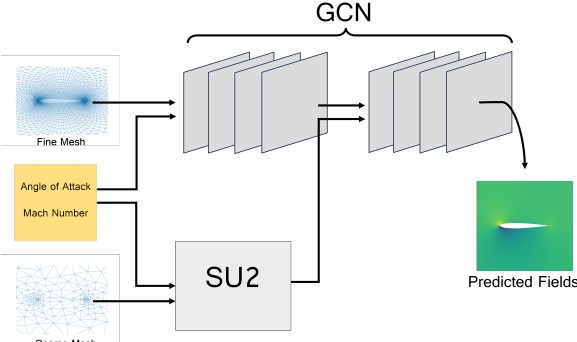

Figure 1: Illustration of CFD-GCN (Belbute-Peres et al., 2020). Both the GCN model parameters and the coarse mesh's node positions are trainable.

Specifically, denote the fine mesh and coarse mesh as $M_{\text{fine}}$ and $M_{\text{coarse}}$, respectively. Each mesh consists of a list of nodes specified by positional $x, y$-coordinates. Now consider a set of $n$ settings of the AoA and Mach number parameters, and denote them as $\{P^1, P^2, ..., P^n\}$. For the $i$-th parameter setting $P^i$, denote the corresponding simulation outputs produced by the SU2 solver with fine and coarse meshes respectively as

$$O^i_{\text{fine}} = \mathbf{Sol}(M_{\text{fine}}, P^i), \tag{1}$$

$$O^i_{\text{coarse}} = \mathbf{Sol}(M_{\text{coarse}}, P^i), \tag{2}$$

where $\mathbf{Sol}(\cdot, \cdot)$ stands for the SU2 solver. The output $O^i_{\text{coarse}}$ is up-sampled to match the fine-mesh resolution through the nearest interpolation algorithm, as described by Belbute-Peres et al. (2020). Notably, there could be infinite ways to up-sample the coarse result; we opt for the specific approach outlined by Belbute-Peres et al. (2020) due to its simplicity and effectiveness in our context. Further denote the graph convolutional

network model as $\text{GCN}_\theta$, where $\theta$ corresponds to the model parameters. Then, the overall training objective function can be written as

$$\min_{\theta, M_{\text{coarse}}} \frac{1}{n} \sum_{i=1}^{n} \mathcal{L}\Big(\text{GCN}_\theta(M_{\text{fine}}, O_{\text{coarse}}^i), O_{\text{fine}}^i\Big), \tag{3}$$

where $\mathcal{L}$ stands for the MSE loss. Here, the goal is to jointly learn and optimize the GCN model parameters $\theta$ and the coordinates of the nodes in the coarse mesh $M_{\text{coarse}}$. The reason is that the coarse mesh usually contains very few number of nodes, and hence the positions of the nodes critically affect the coarse simulation outcome $O_{\text{coarse}}$, which further affects the prediction accuracy of the GCN model.

## 2.3 Differentiability of Hybrid Model

To solve the above optimization problem, one standard approach is to use gradient-based methods such as SGD, which require computing the gradient $[\frac{\partial \mathcal{L}}{\partial \theta}; \frac{\partial \mathcal{L}}{\partial M_{\text{coarse}}}]$ using stochastic samples via back-propagation. Specifically, the partial derivative $\frac{\partial \mathcal{L}}{\partial \theta}$ can be calculated by standard machine learning packages that support automatic differentiation such as PyTorch (Paszke et al., 2019) and TensorFlow (Abadi et al., 2016). For the other partial derivative $\frac{\partial \mathcal{L}}{\partial M_{\text{coarse}}}$, it can be decomposed as follows by the chain rule.

$$\frac{\partial \mathcal{L}}{\partial M_{\text{coarse}}} = \frac{\partial \mathcal{L}}{\partial O_{\text{coarse}}} \cdot \frac{\partial O_{\text{coarse}}}{\partial M_{\text{coarse}}}. \tag{4}$$

Note that the term $\frac{\partial \mathcal{L}}{\partial O_{\text{coarse}}}$ can also be calculated by standard machine learning packages. The challenge is to compute the other term $\frac{\partial O_{\text{coarse}}}{\partial M_{\text{coarse}}}$, which corresponds to the derivative of the output of the solver with regard to the input coarse mesh. In Belbute-Peres et al. (2020), the authors proposed to compute this term by adopting the SU2 solver that supports automatic differentiation. However, this requirement can be restrictive for general black-box solver. Notably, in physics and chemistry disciplines, ML models may be required to interface with experiments or complex simulation codes for which the underlying systems are black-box (Thelen et al., 2022; Tsaknakis et al., 2022; Louppe et al., 2019; Abreu de Souza et al., 2023; Baydin et al., 2020).

## 3 Differentiating Numerical Solver via Forward Propagation

In scenarios where the PDE solver does not support automatic differentiation, we propose to estimate the partial derivative $\frac{\partial O_{\text{coarse}}}{\partial M_{\text{coarse}}}$ by leveraging the simulation outcomes, i.e., forward propagation through the solver. Fortunately, the classic zeroth-order gradient estimators provide solutions to estimate high-dimensional gradients using only forward propagation (Liu et al., 2018a). Next, we apply the classic coordinate-wise and Gaussian formulas of the zeroth-order gradient estimators to estimate the derivative of the solver. To simplify the presentation, we focus on scalar-valued simulation outcome $O$ as a function of the input mesh $M$. In practice, $O$ is vector-valued and we simply apply the estimators to each dimension of it.

### 3.1 Coordinate-wise Gradient Estimator

The coordinate-wise zeroth-order gradient estimator estimates the gradient of the solver via the following steps. First, we sample a mini-batch $b \in \mathbb{N}$ of the coordinates of the input coarse mesh nodes uniformly at random. In our setting, each node has two coordinates to specify its position in the $x$-$y$ plane. We denote the index of these sampled coordinates as $\{\xi_1, \xi_2, ..., \xi_b\}$, and denote the Euclidean basis vector associated with the coordinate $\xi_j$ as $e_{\xi_j}$. Then, choose parameter $\mu > 0$, the coordinate-wise zeroth-order (Coordinate-ZO) gradient estimator is constructed as follows.

$$\text{(Coordinate-ZO):} \quad \frac{\widehat{\partial O}}{\partial M} := \frac{1}{b} \sum_{j=1}^{b} \frac{O(M + \mu e_{\xi_j}) - O(M)}{\mu} e_{\xi_j}. \tag{5}$$

To elaborate, Coordinate-ZO estimates the gradient based on the finite-difference formula, which requires to perturb the input mesh $M$ over the sampled coordinates $\{e_{\xi_j}\}_{j=1}^{b}$ and compute their corresponding simulation outcomes via running the solver.

**Remark.** *The main advantage of using the Coordinate-ZO estimator is its high asymptotic accuracy as $\mu$ tends to 0. When $\mu \downarrow 0$, each term in the above summation converges to the exact partial gradient of $O$ with regard to the corresponding node coordinate. In practice, however, setting $\mu$ too small can lead to numerical instability and less accurate evaluations. The bias due to the a non-vanishing $\mu$ often results in subpar performance compared to automatic differentiation (AD).*

*Furthermore, comparing to the finite-difference method, Coordinate-ZO requires only $b$ function evaluations per mesh update step. For instance, a forward pass in SU2 takes approximately 2.0 seconds on the coarse mesh (Belbute-Peres et al., 2020), leading to around 1000 seconds for a full finite-difference step, while the Coordinate-ZO estimation takes only $2.0 \times b$ seconds. Due to such high time cost, we did not provide the necessary rounds of SU2 simulations of obtaining the required accuracy; instead, we trade-off between the function call complexity and the model accuracy by tuning the parameter $b$.*

### 3.2 Gaussian Gradient Estimator

Another popular zeroth-order gradient estimator estimates the gradient via Gaussian perturbations. Specifically, we first generate a mini-batch $b \in \mathbb{N}$ of standard Gaussian random vectors, each with dimension $d$ that equals 2x of the number of nodes in the mesh (because each node has two coordinates). Denote these generated Gaussian random vectors as $\{g_1, g_2, ..., g_b\} \sim \mathcal{N}(0, I)$. Then, choose parameter $\mu > 0$, the Gaussian zeroth-order (Gaussian-ZO) gradient is constructed as follows.

$$\text{(Gaussian-ZO):} \quad \frac{\widehat{\partial O}}{\partial M} := \frac{1}{b} \sum_{j=1}^{b} \frac{O(M + \mu g_j) - O(M)}{\mu} g_j. \tag{6}$$

Note that the Gaussian-ZO estimator perturbs the mesh using a dense Gaussian vector $g_j$. This is very different from the Coordinate-ZO that perturbs a single coordinate of the mesh nodes using $e_{\xi_j}$. In fact, Gaussian-ZO is a stochastic approximation of the gradient of a Gaussian-smoothed objective function $O_\mu(M) := \mathbb{E}_g[O(M + \mu g)]$.

**Remark.** *The main advantage of using the Gaussian-ZO estimator is that it estimates the gradient over the full dimensions. Therefore, even with batch size $b = 1$, the formula still generates a gradient estimate over all the coordinates. This is an appealing property to the CFD-GCN learning system, as one can update the entire coarse mesh with just one Gaussian noise (one additional query of simulation). However, the gradient estimate generated by Gaussian-ZO usually suffers from a high variance, and hence a large batch size is often needed to control the variance. Moreover, the Gaussian-ZO formula estimates the gradient of the smoothed objective function $O_\mu(M)$, which introduces extra approximation error.*

### 3.3 Gaussian-Coordinate Gradient Estimator

To avoid the high variance issue of the Gaussian-ZO estimator and the low sample efficiency issue of the Coordinate-ZO estimator, we propose Gaussian-Coordinate-ZO estimator – a mixed zeroth-order gradient estimator that leverages the advantages of both estimators. Specifically, we first randomly sample a mini-batch of $d$ coordinates of the mesh nodes, denoted as $D := \{\xi_1, \xi_2, ..., \xi_d\}$. Then, we generate a mini-batch of $b$ standard Gaussian random vectors, denoted as $\{g_1, g_2, ..., g_b\}$. Lastly, we construct the Gaussian-Coordinate-ZO estimator as follows with parameter $\mu > 0$, where $[g]_D$ denotes a vector whose coordinates excluding $D$ are set to be zero.

$$\text{(Gaussian-Coordinate-ZO):} \quad \frac{\widehat{\partial O}}{\partial M} := \frac{1}{b} \sum_{j=1}^{b} \frac{O(M + \mu[g_j]_D) - O(M)}{\mu} [g_j]_D. \tag{7}$$

Intuitively, the Gaussian-Coordinate-ZO estimator can be viewed as a Gaussian-ZO estimator applied only to the coordinates in $D$.

**Remark.** *Compared to the Gaussian-ZO estimator that estimates the gradient over all the coordinates, Gaussian-Coordinate-ZO estimates the gradient only over the coordinates in $D$, and hence is less noisy. On the other hand, unlike the Coordinate-ZO estimator that estimates the gradient over a single coordinate per*

*sample, the Gaussian-Coordinate-ZO estimator estimates the gradient over a mini-batch of coordinates in D even with a single Gaussian sample. This leads to an improved sample/learning efficiency as validated by our experiments later.*

## 4 Experiments

### 4.1 Experiment Setup

We apply the three aforementioned zero-order gradient estimators with parameter $\mu = 1e-3$ to estimate the gradient of the PDE solver output with regard to the input coarse mesh coordinates, and then use the full gradient $[\frac{\partial \mathcal{L}}{\partial \theta}; \frac{\partial \mathcal{L}}{\partial M_{\text{coarse}}}]$ to train the model parameters $\theta$ and the coarse mesh coordinates $M_{\text{coarse}}$ by optimizing the objective function in eq. (3). The training dataset and test dataset consist of the following range of AoA and mach numbers.

Table 1: List of training data and test data.

|  | AoA | Mach Number |
| --- | --- | --- |
| Training data | [-10:1:10] | [0.2:0.05:0.45] |
| Test data | [-10:1:10] | [0.5:0.05:0.7] |

The fixed fine mesh $M_{\text{fine}}$ contains 6648 nodes, and the trainable coarse mesh $M_{\text{coarse}}$ contains 354 nodes that are initialized by down-sampling the fine mesh. Moreover, for the training, We use the standard Adam optimizer (Kingma & Ba, 2014) with learning rate $5 \times 10^{-5}$ and batch size 16 (for sampling the AoA and mach number).

We compare our ZO methods with two baselines: (i) the gradient-based approach (referred to as *Grad*) proposed in Belbute-Peres et al. (2020), which requires a differentiable PDE solver; and (ii) the gradient-based approach but with a frozen coarse mesh (referred to as *Grad-FrozenMesh*), which does not optimize the coarse mesh at all.

### 4.2 Results and Discussion

**1. Coordinate-ZO.** We first implement the Coordinate-ZO approach with different batch sizes $b = 1, 2, 4$ for sampling the node coordinates (see eq. (5)), and compare its test loss with that of the two gradient-based baselines. Figure 2 (left) shows the obtained test loss curves over the training epochs. It can be seen that the test loss curves of Coordinate-ZO are lower than that of the Grad-FrozenMesh approach and are higher than that of the Grad approach. This indicates that optimizing the coarse mesh using the Coordinate-ZO estimator leads to a lower test loss than using a frozen coarse mesh, and leads to a higher test loss than gradient-based mesh optimization. This is expected as the gradient estimated by the Coordinate-ZO estimator is in general sparse and noisy, which slows down the convergence and degrades the test performance. In particular, as the batch size $b$ increases, Coordinate-ZO achieves a lower test loss at the cost of running more coarse mesh simulations, as illustrated by Figure 2 (right). We note that the Grad-FrozenMesh approach does not update the coarse mesh at all and hence is not included in the right figure.

The following Figure 3 visualizes the pressure fields predicted by the Grad, Grad-FrozenMesh baselines and our Coordinate-ZO approach with batch size $b = 4$, for input parameters AoA = 9.0 and mach number = 0.8. It can be seen that the field predicted by our Coordinate-ZO approach is more accurate than that predicted by the Grad-FrozenMesh baseline, due to the optimized coarse mesh. Also, the highlighted yellow region of the field predicted by our approach looks very close to that predicted by the Grad baseline (which uses a differentiable solver).

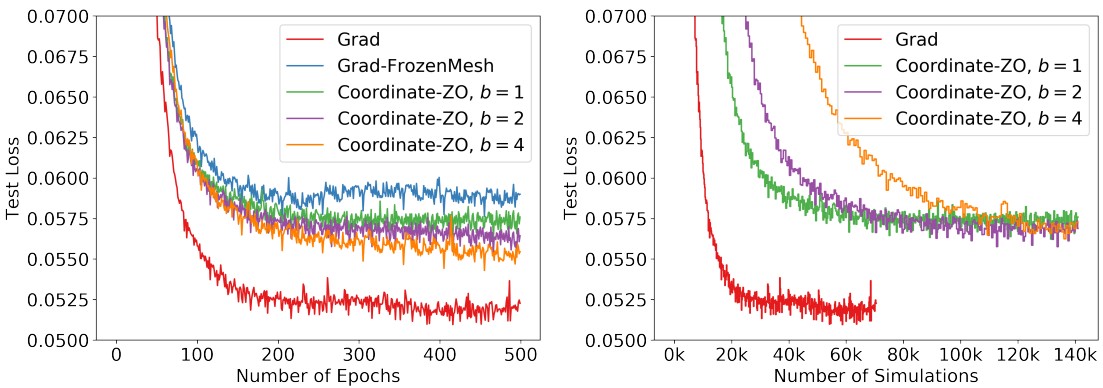

Figure 2: Test loss comparison among Coordinate-ZO, Grad and Grad-FrozenMesh.

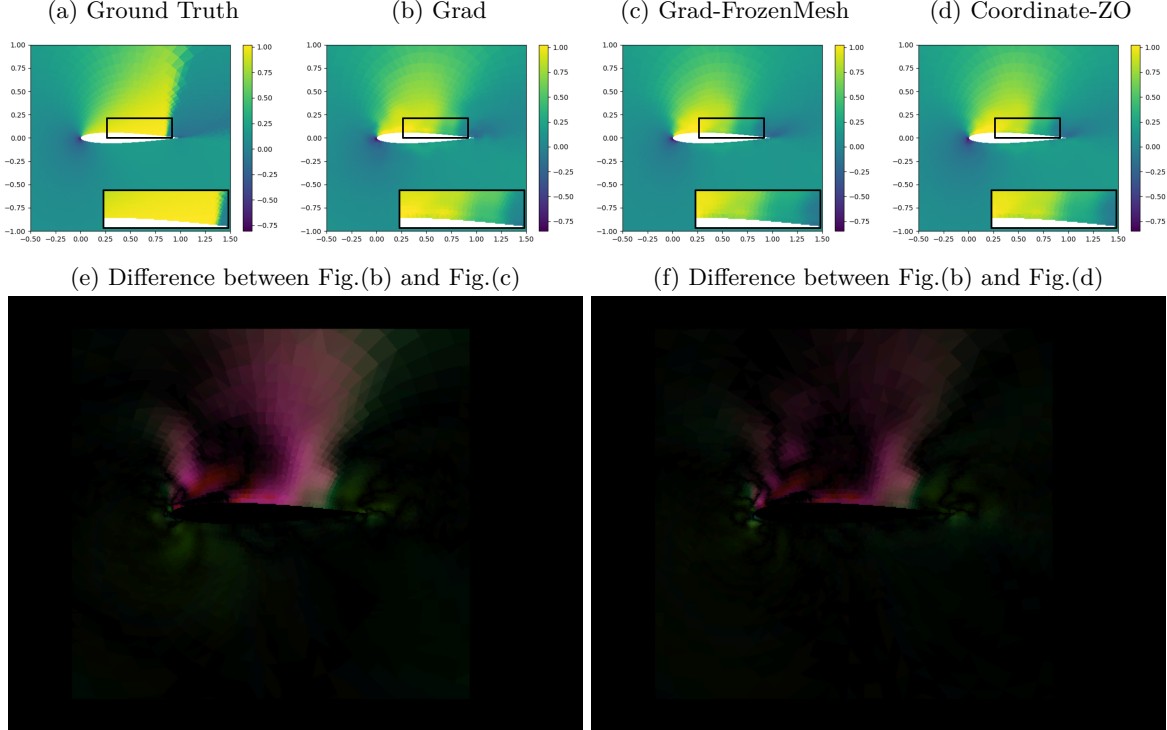

Figure 3: Visualization of the pressure fields predicted by Grad, Grad-FrozenMesh, and Coordinate-ZO with $b = 4$ for AoA = 9.0 and mach number = 0.8. We further evaluate the difference between the prediction over meshes updated by first-order and Coordinate-ZO method in Fig.(e) and Fig.(f). Notably, exhibits a significantly higher light intensity than Fig. (f), indicating a larger divergence between the figures. This observation implies that even when updating the mesh is not applicable (e.g. the auto-differentiation is not supported), we can still apply the zeroth-order method (Coordinate-ZO) to update these parameters and obtain more consistent result as it does (Grad). We have adjusted the brightness and contrast to better distinguish the difference.

Figure 4 compares the optimized coarse mesh (red) obtained by our Coordinate-ZO approach with the original frozen coarse mesh (blue). It can be seen that many nodes' positions have been updated by Coordinate-ZO to improve the overall prediction performance. In particular, the nodes in the dense area tend to have a

more significant shift than those in the sparse area. This is reasonable as the dense nodes typically play a more important role in simulations.

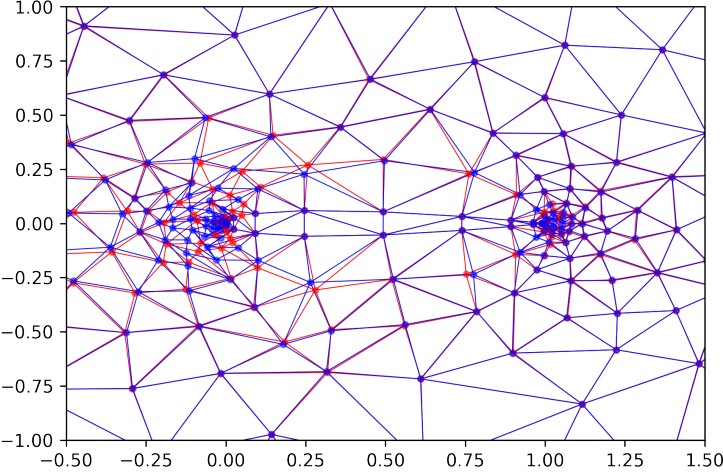

Figure 4: Comparison between the mesh before (blue) and after (red) being optimized.

**2. Gaussian-ZO.** We further implement the Gaussian-ZO approach with different batch sizes $b = 1, 2, 4$ and compare its test loss with that of the two baselines. Figure 5 shows the obtained comparison results, which are very similar to those obtained by Coordinate-ZO in Figure 2. This indicates that Gaussian-ZO can optimize the coarse mesh with a comparable performance to that of the Coordinate-ZO approach, and the convergence speed is slower than the gradient-based Grad approach due to the intrinsic high variance of the Gaussian-ZO estimator.

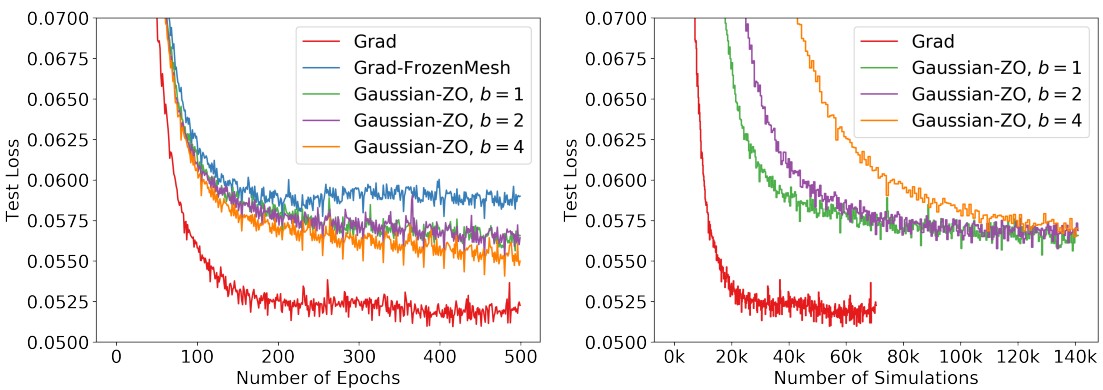

Figure 5: Test loss comparison among Gaussian-ZO, Grad, and Grad-FrozenMesh.

The following Figure 6 visualizes the pressure fields predicted by the Grad, Grad-FrozenMesh baselines and our Gaussian-ZO approach with batch size $b = 4$, for input parameters AoA $= -10.0$ and Mach $= 0.8$. One can see that the highlighted yellow area predicted by Gaussian-ZO is more accurate than that predicted by Grad-FrozenMesh, and is close to the predictions of the Grad approach.

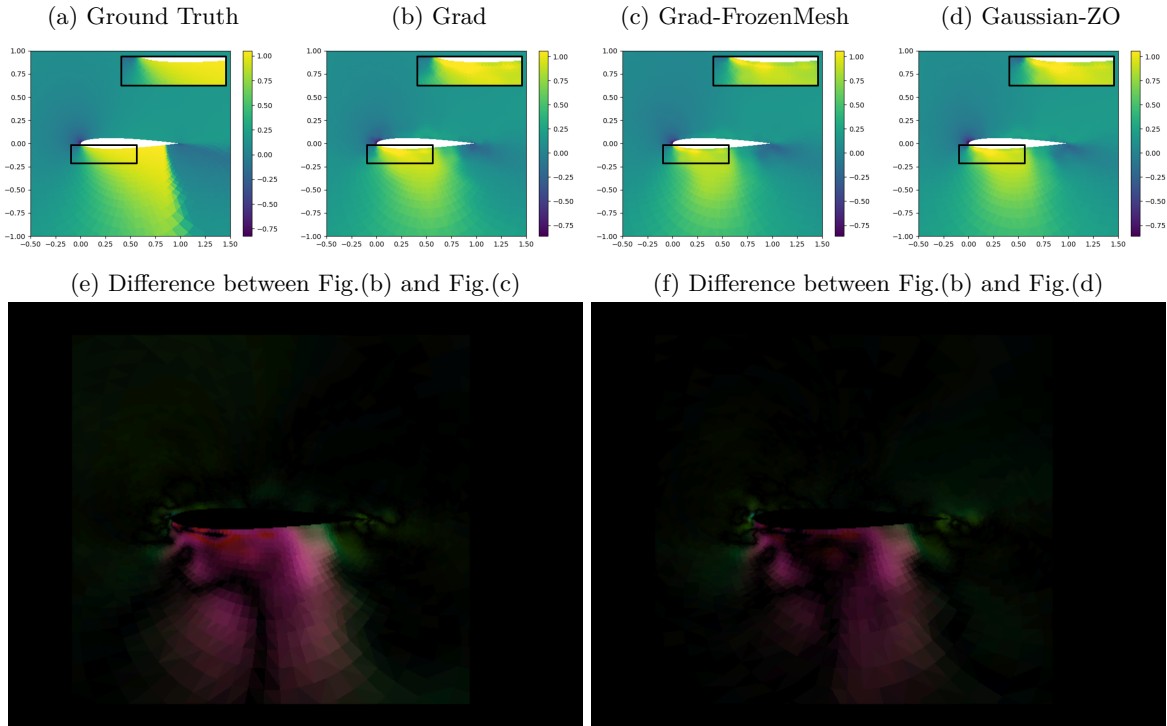

(a) Ground Truth  (b) Grad  (c) Grad-FrozenMesh  (d) Gaussian-ZO

(e) Difference between Fig.(b) and Fig.(c)   (f) Difference between Fig.(b) and Fig.(d)

Figure 6: Compare simulation results with Grad, Grad-FrozenMesh and Gaussian-ZO with $b = 4$ for AoA = $-10.0$ and Mach = 0.8. We further evaluate the difference between the prediction over meshes updated by first-order and Gaussian-ZO method in Fig.(e) and Fig.(f). Here, higher light intensity indicates a larger divergence between the figures. This underscores the advantage of our proposed zeroth-order optimization framework, as it ensures compatibility in contexts where first-order methods are unfeasible.

**3. Gaussian-Coordinate-ZO.** Lastly, we test the Gaussian-Coordinate-ZO approach (referred to as *Gauss-Coord-ZO*) with different choices of parameters $d$ and $b$ (see eq. (7)). Figure 7 (left) shows the comparison result under a fixed batch size $b = 1$ and different choices of $d$. Due to the fixed batch size, these Gauss-Coord-ZO algorithms query the same total number of simulations, and one can see that increasing $d$ leads to a slightly lower test loss. Moreover, Figure 7 (right) shows the comparison result under a fixed $d = 4$ and different choices of $b$. One can see that increasing $b$ leads to a lower test loss due to the reduced variance of the Gaussian-Coordinate-ZO estimator. Overall, we found that the choice of parameters $b = 1, d = 16$ achieves the best balance between performance and sample/time efficiency.

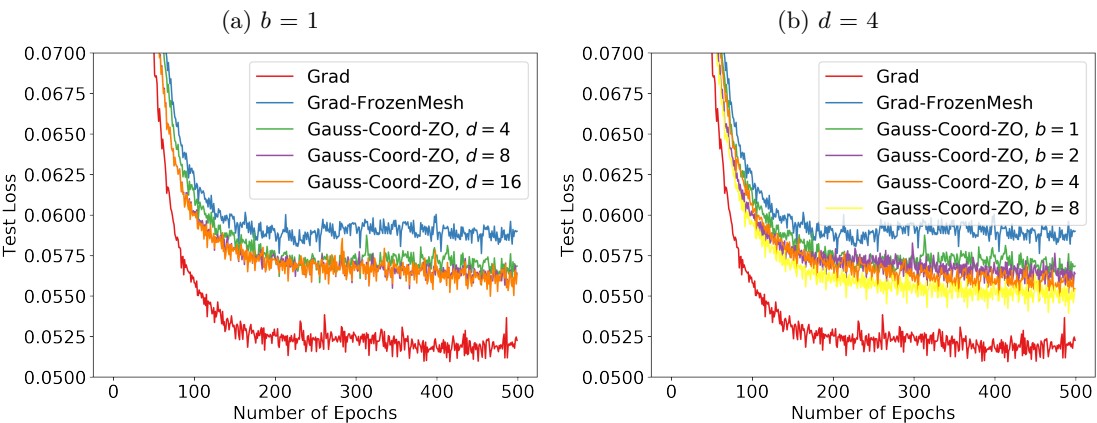

Figure 7: Test loss comparison of Gauss-Coord-ZO with different choices of $b$ and $d$.

We further compare the performance of Gauss-Coord-ZO ($d = 16$) with that of the two baselines and the previous two zeroth-order approaches. Figure 8 shows the comparison results obtained with different choices of batch size. With batch size $b = 1$, it can be seen from the left figure that Gauss-Coord-ZO achieves a slightly lower test loss than Coordinate-ZO and Gaussian-ZO, demonstrating the advantage of this mixed zeroth-order estimator. On the other hand, as the batch size increases to $b = 4$ in the right figure, all the three zeroth-order approaches achieve a similar test loss, and is slightly better than that achieved by Gauss-Coord-ZO with $b = 1$. Overall, we observe that the Gauss-Coord-ZO estimator with $b = 1, d = 16$ already leads to a good practical performance. This choice of parameter is also favorable as the batch size $b = 1$ requires running the minimum number of simulations. Notably, all three zeroth-order approaches do not match the first-order baseline. According to Nesterov & Spokoiny (2017), the efficacy of zeroth-order optimization is limited by the dimensionality of the input. To achieve comparable accuracy within the same number of epochs, it is necessary to adjust the batch size to approximately $352 \times 2$, which is a fundamental limitation of zeroth-order techniques. Despite this, our proposed framework presents a viable alternative for optimizing parameters in external PDE solvers that lack auto-differentiation capabilities.

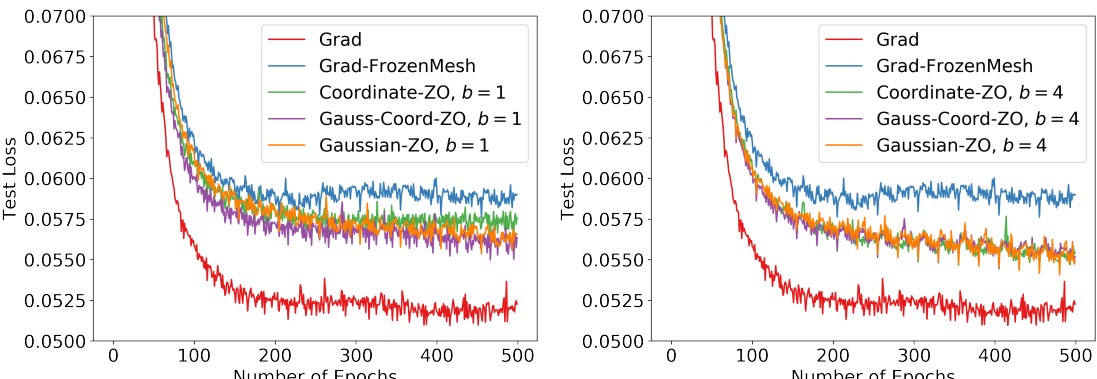

Figure 8: Test loss comparison among Coordinate-ZO, Gaussian-ZO, and Gauss-Coord-ZO.

Figure 9 shows the comparison among the simulation results predicted by Grad, Grad-FrozenMesh and Gaussian-Coordinate-ZO, all with batch size $b = 4$ and input parameters AoA = 5.0 and Mach = 0.4. It can be seen that all the three zeroth-order approaches obtain very similar prediction results that are close to the ground truth.

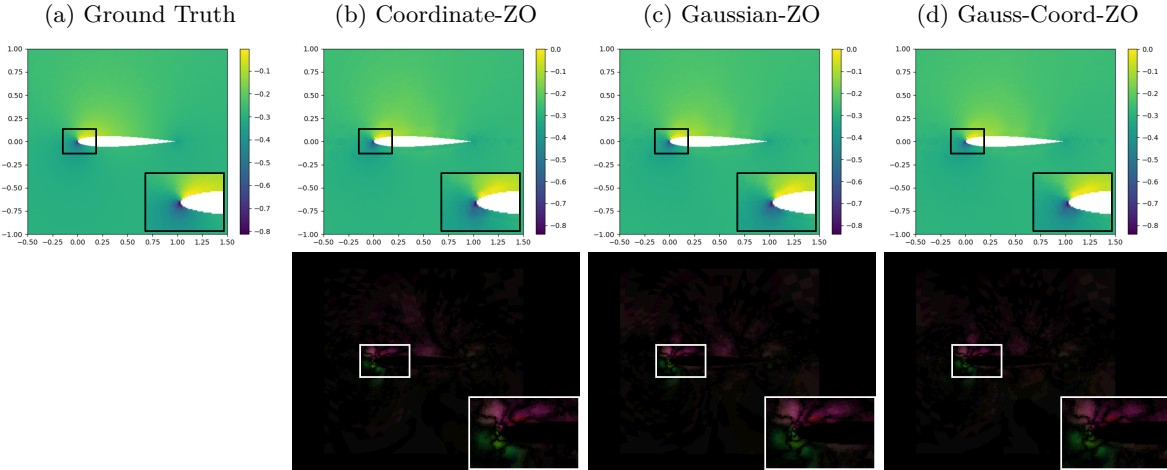

Figure 9: Compare simulation results with Coordinate-ZO, Gaussian-ZO and Gauss-Coord-ZO with $b = 4$ for AoA = 5.0 and Mach = 0.4. The comparison reveals that Gauss-Coord-ZO most closely approximates the ground truth, as evidenced by its difference figures being darker than those produced by the other two methods.

## 4.3 Improve Generalizability via Warm-Start

In this subsection, we propose a warm-start strategy to further improve the generalization performance of the proposed zeroth-order approaches.

**Asymmetry in Optimization.** One key observation is that the optimization of the objective function in eq. (3) is highly asymmetric between the network model parameters $\theta$ and the coarse mesh $M_{\text{coarse}}$. To elaborate, optimizing deep neural network's model parameters is known to be a challenging problem due to nonconvex optimization, and the trained parameters are often substantially different from the initialized parameters. As a comparison, optimizing coarse mesh is far less complex, since the optimized mesh is usually close to the initialized mesh, as shown in Figure 4. Thus, in the initial training phase when the neural network parameters' gradients are large and noisy, they tend to amplify the noise of the mesh nodes' estimated gradients through the chain rule in eq. (4), which further slows down the overall convergence and degrades the generalization performance.

**Warm-Start.** With the above insight, we propose a simple warm-start strategy to accelerate the training of the hybrid model using zeroth-order approaches and improve the generalization performance. Specifically, we propose the following two-stage training process.

- Warm-up Stage: we first freeze the coarse mesh and train only the neural network parameters for 300 epochs. We note that this stage does not need to query any additional simulations due to the frozen coarse mesh.

- Stage two: with the model obtained in the previous stage, we unfreeze the coarse mesh and jointly train it with the network parameters using any of the previously discussed three zeroth-order algorithms.

Table 2: Comparison of the best test loss achieved by the zeroth-order approaches within 200 epochs using random initialization and warm-start. All approaches query the same number of simulations, and the batch size $b$ is set to be 1.

|            | Gauss-Coord-ZO | Coord-ZO | Gauss-ZO |
|------------|----------------|----------|----------|
| Random     | 0.05650        | 0.05721  | 0.05745  |
| Warm-start | 0.05540        | 0.05463  | 0.05455  |

Table 2 compares the best test loss achieved by all the three zeroth-order approaches with $b = 1$ within 200 training epochs using random initialization and warm-start. Note that under both random initialization and warm-start, all the approaches query the same total number of simulations. It can be seen that with the warm-start strategy, all the zeroth-order approaches achieve improved generalization performance without querying additional simulations.

Figure 10 further compares the predicted pressure fields generated by these zeroth-order approaches under random initialization and warm-start for input parameters AoA = 5.0 and Mach = 0.8. It can be seen that all the simulation results produced by the model trained with warm-start are more accurate than those trained with random initialization.

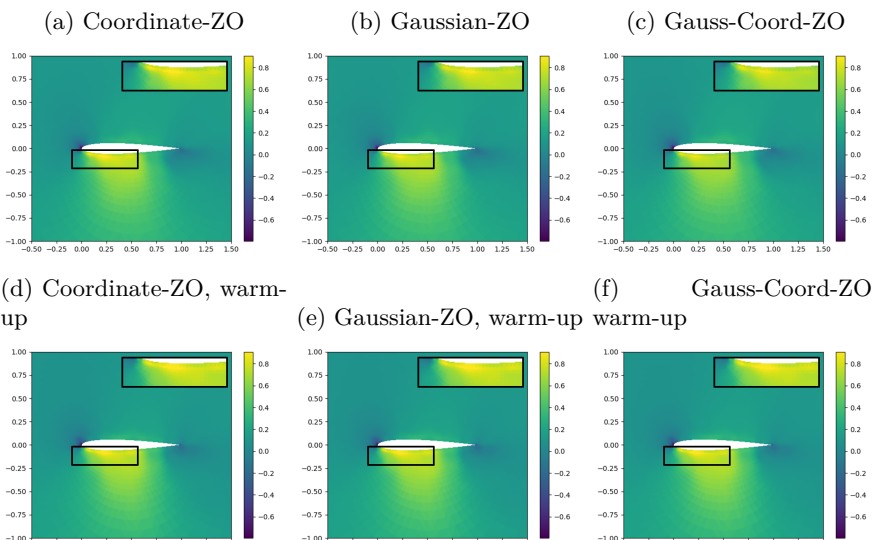

Figure 10: Comparison of simulation results predicted by training with random initialization and warm-start.

## 5   Conclusion

This study developed a learning system for fluid flow prediction that supports an end-to-end training of black-box PDE solvers and deep learning models. We investigated the performance of optimizing this system using various zeroth-order estimators, which allow us to differentiate the solver without querying the exact gradients. Experiments showed that our approaches have competitive performance compare to gradient-based baselines, especially when adopting a warm-start strategy. We expect that our research will help integrate physical science modules into modern deep learning without the need for substantial adaptation.

**Broader Impact Statement**

The paper introduces a novel approach for integrating black-box Partial Differential Equation (PDE) solvers with deep learning models in computational fluid dynamics (CFD). It focuses on training a hybrid model that

combines an external PDE solver and a deep graph neural network, using zeroth-order gradient estimators to optimize the solver's mesh parameters and first-order gradient to optimize the neural network's parameters. This methodology addresses the challenge of integrating traditional CFD solvers, which often lack automatic differentiation capabilities, with modern deep learning techniques. This advancement has the potential to significantly enhance the accuracy and efficiency of fluid flow predictions and can be extended to various fields where black-box simulators play a crucial role, offering new opportunities for interdisciplinary research and innovation.

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

## A   Compare three zeroth-order methods in the same number epochs

In Figure 11, we present the comparison among three zeroth-order methods under different batch size $b$. The dimension $d$ of the Gauusian-Coordinate-ZO method is fixed to be 16. The Figure 11-(a) and Figure 11-(c) have been detailed in the main text.

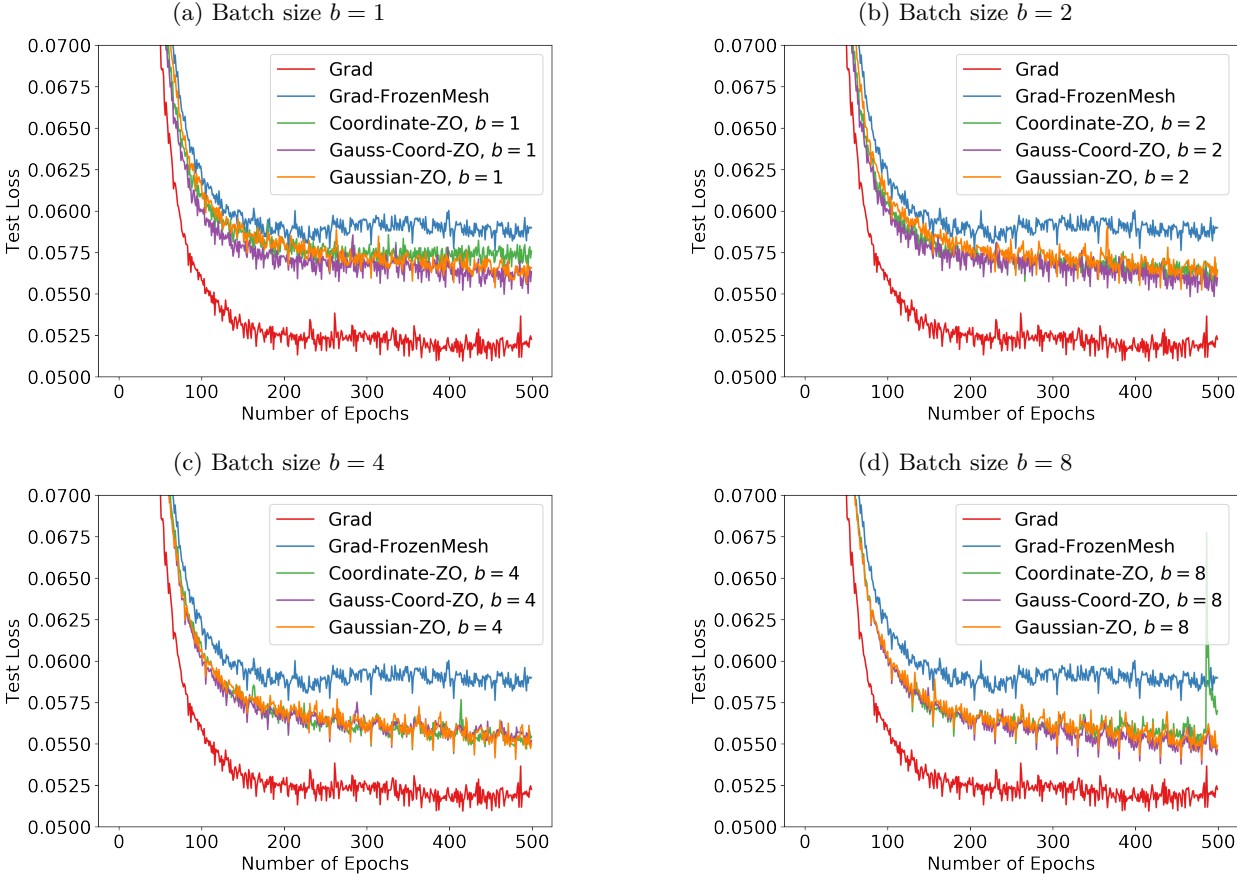

Figure 11: A comparison of three zeroth-order methods across varying batch sizes $b$. Gaussian-Coordinate-ZO typically outperforms the others at smaller batch sizes and matches their performance as the batch size increases.

# B  Loss curves for Coordinate-Gaussian-ZO in number of epochs

To further examine the impact of the hyper-parameters on Coordinate-Gaussian-ZO, we compared various hyper-parameters. In Figure 12, we keep the batch size $b$ constant while increasing the dimension $d$. Conversely, in Figure 13, the dimension $d$ was held steady as we increase the batch size $b$.

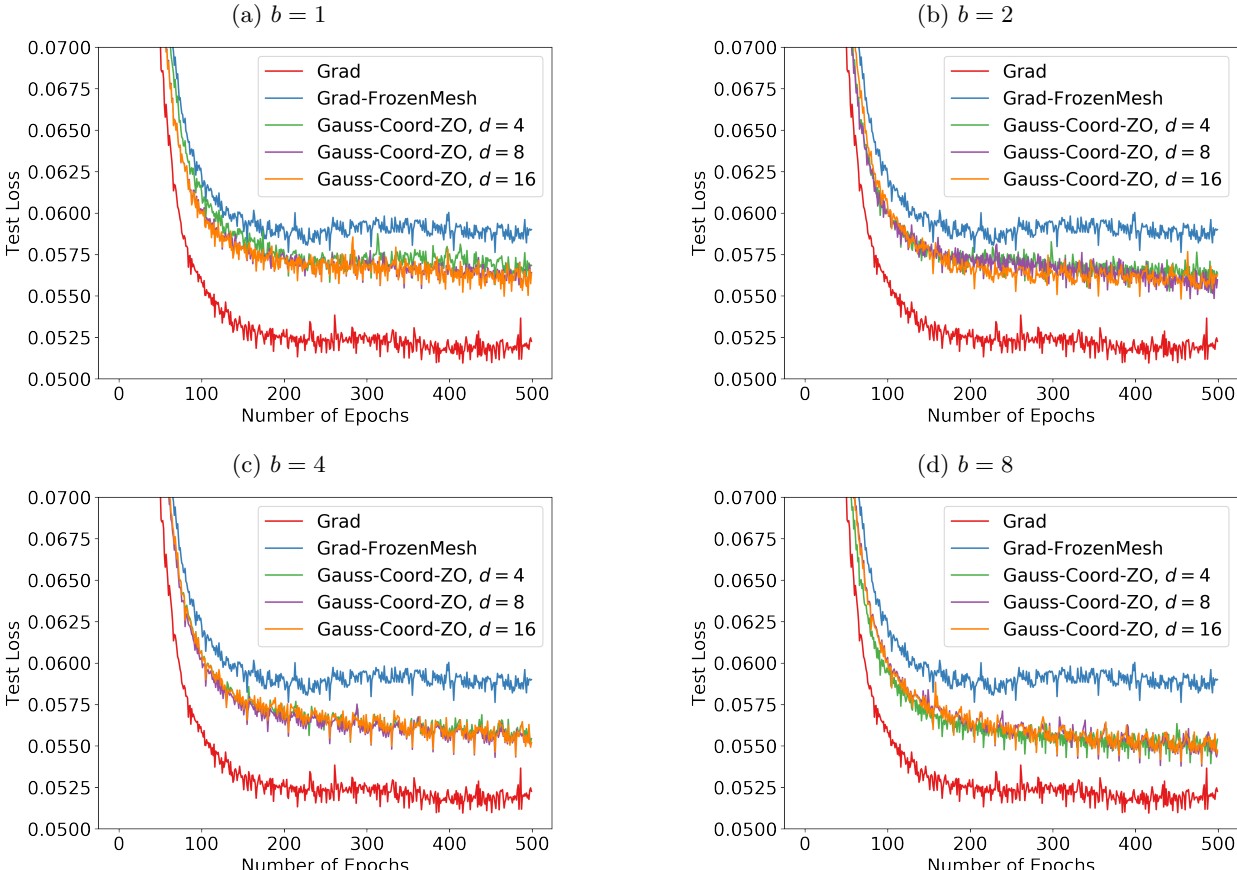

Figure 12: Illustration of the impact of increasing dimension $d$ with constant batch size $b$. Greater performance gains are observed when increasing $d$ with a smaller batch size such as $b = 1$ or $b = 2$. However, there is no further improvement when $d$ is over 8.

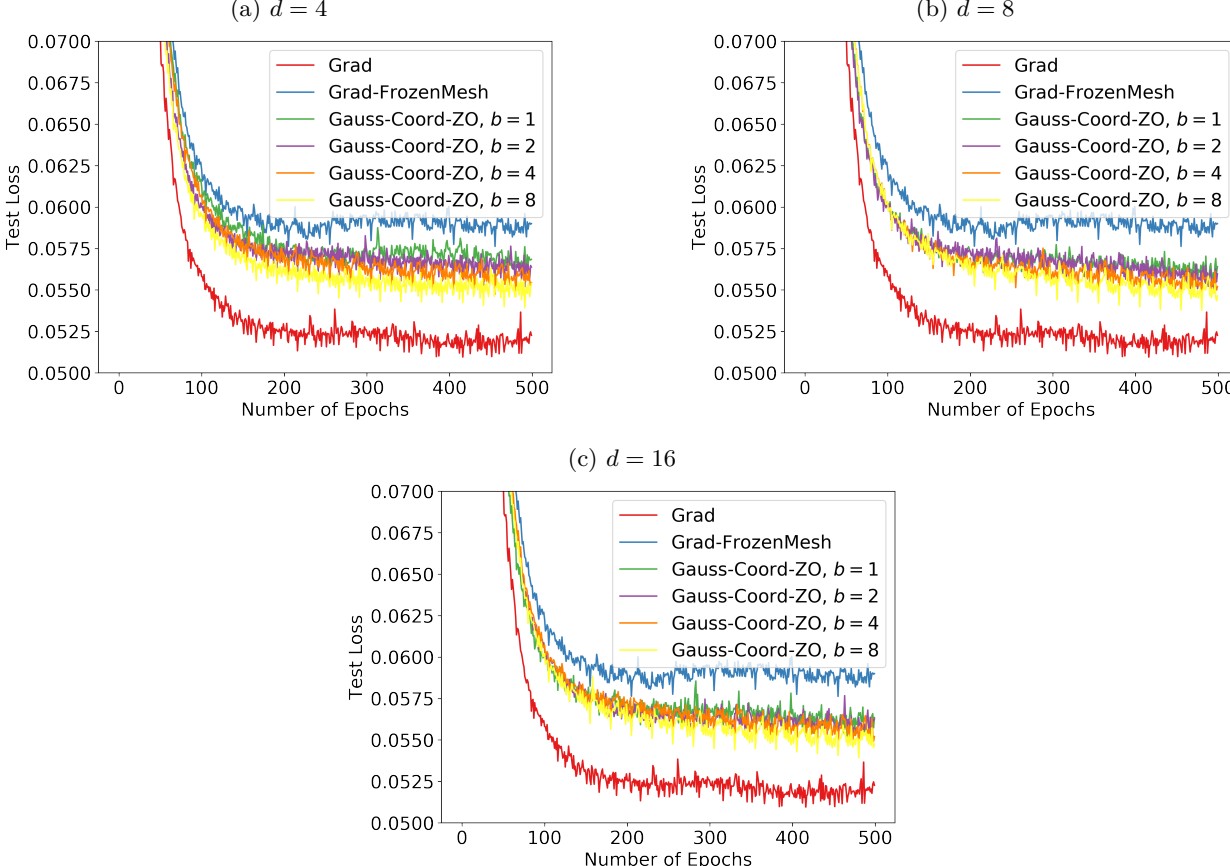

Figure 13: Illustration of the impact of increasing batch size $b$ with constant dimension $d$. The generalization performance of Gaussian-Coordinate-ZO is always improved with a larger batch size.

## C  Simulations

This section provides an in-depth look at the simulation results for two different pair of AoA and Mach number under different predicted fields. Specifically, we have selected two distinct pairs of Angle of Attack (AoA) and Mach number as representative scenarios to illustrate predicted fields from the partially-differentiable hybrid model. For each of comparison, results are presented across the ground truth, Grad-FrozenMesh, Grad, and the corresponding zeroth-order method.

## C.1 Coordinate-ZO

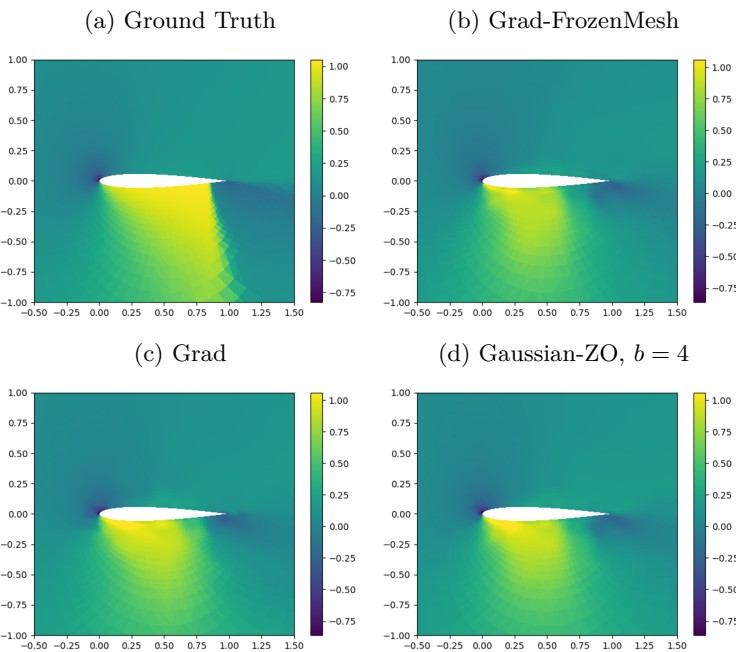

Figure 14: Predicted pressure fields for NACA0012 airfoil under AoA = −10.0 and mach number = 0.8.

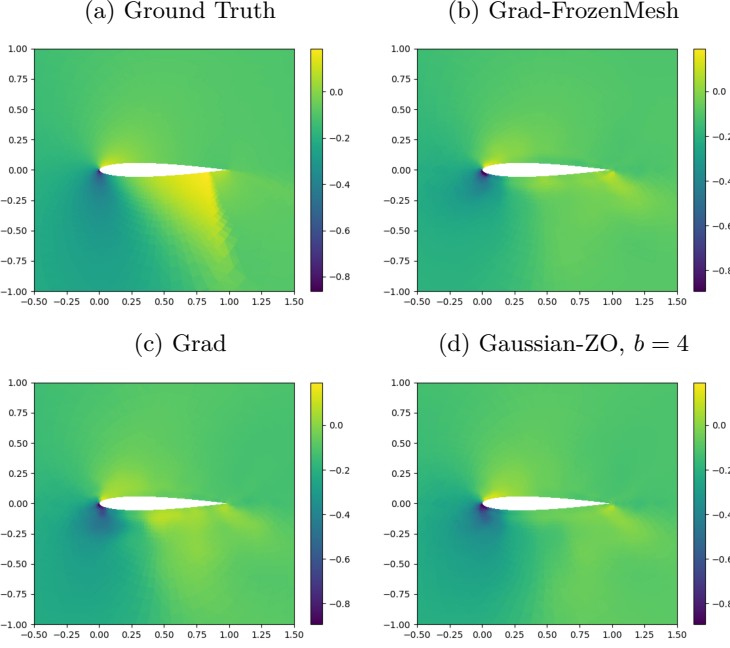

Figure 15: Predicted X-velocity fields for NACA0012 airfoil under AoA = −10.0 and mach number = 0.8.

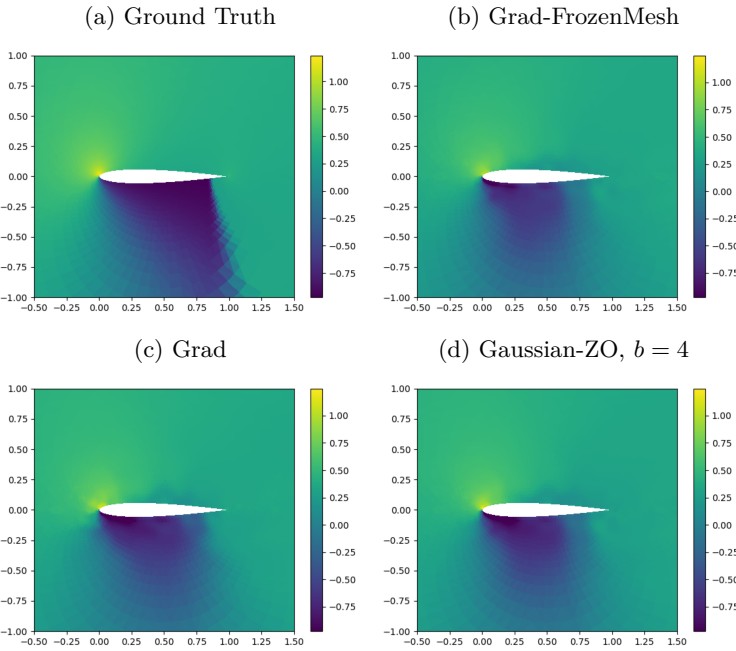

Figure 16: Predicted Y-velocity fields for NACA0012 airfoil under AoA = −10.0 and mach number = 0.8.

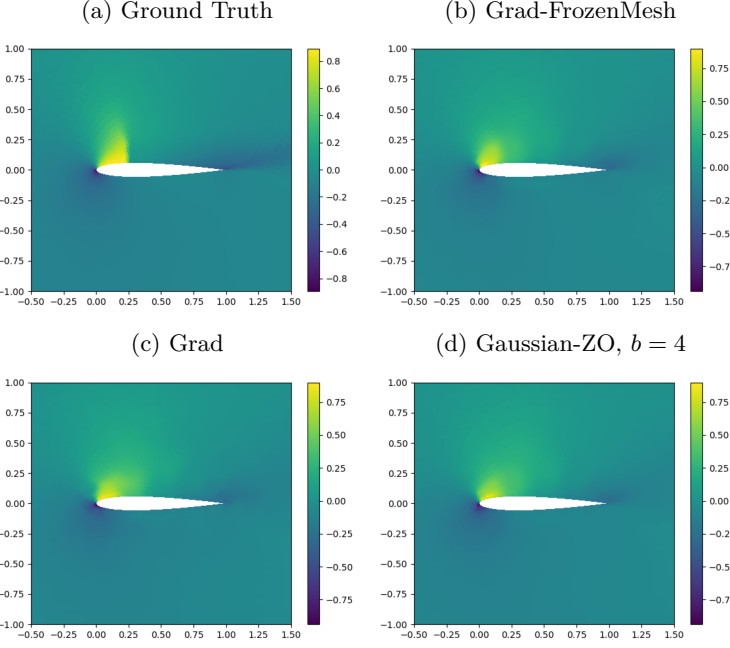

Figure 17: Predicted pressure fields for NACA0012 airfoil under AoA = 9.0 and mach number = 0.6.

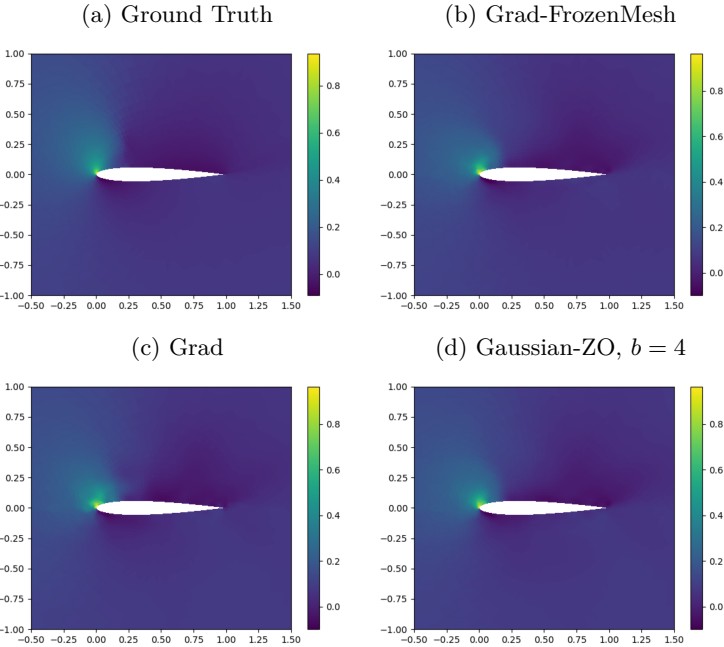

Figure 18: Predicted X-velocity fields for NACA0012 airfoil under AoA = −9.0 and mach number = 0.6.

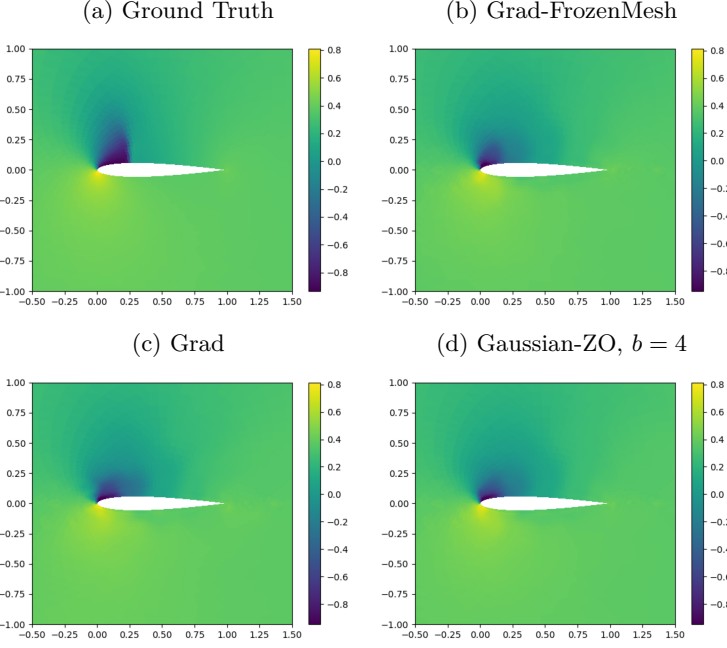

Figure 19: Predicted Y-velocity fields for NACA0012 airfoil under AoA = −9.0 and mach number = 0.6.

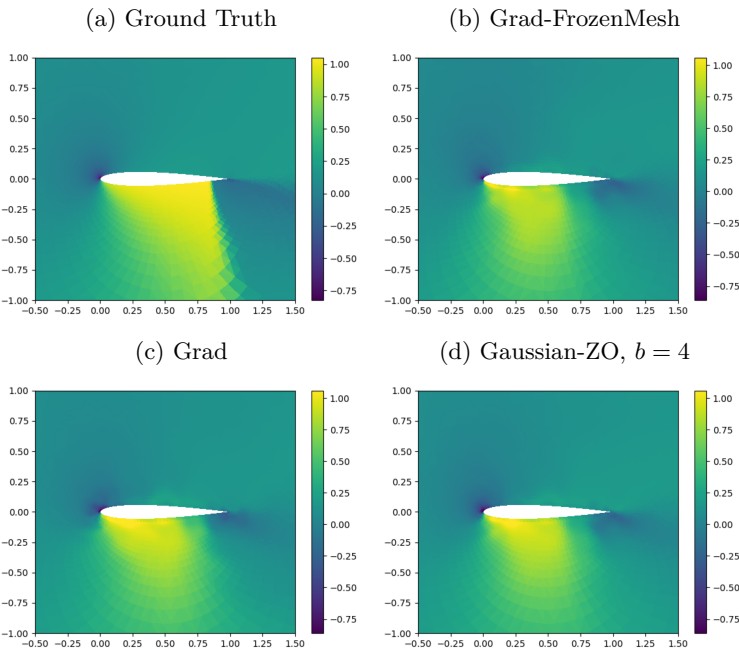

Figure 20: Predicted pressure fields for NACA0012 airfoil under AoA $= -10.0$ and mach number $= 0.8$.

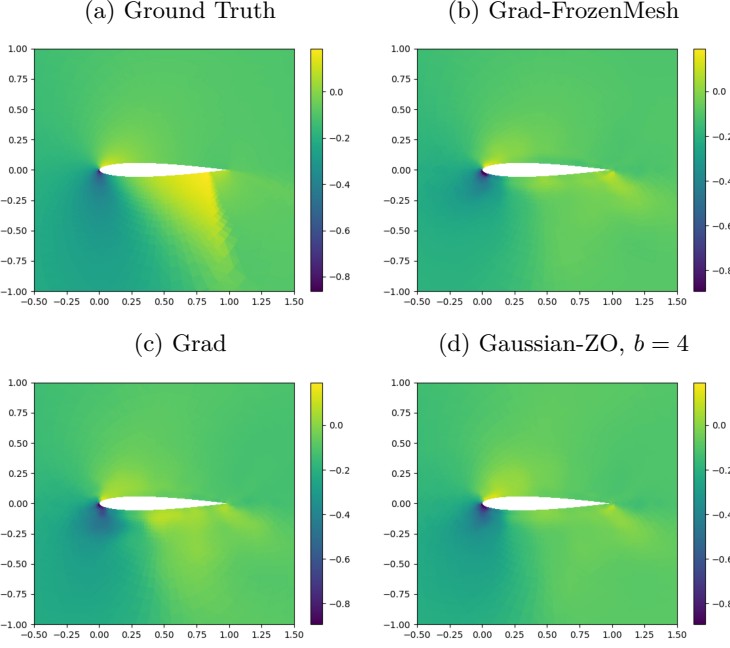

Figure 21: Predicted X-velocity fields for NACA0012 airfoil under AoA $= -10.0$ and mach number $= 0.8$.

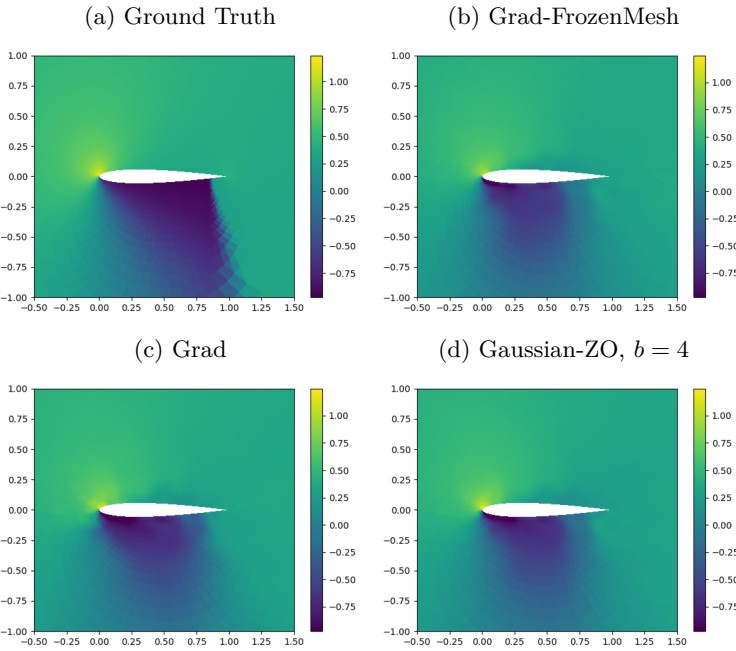

Figure 22: Predicted Y-velocity fields for NACA0012 airfoil under AoA = −10.0 and mach number = 0.8.

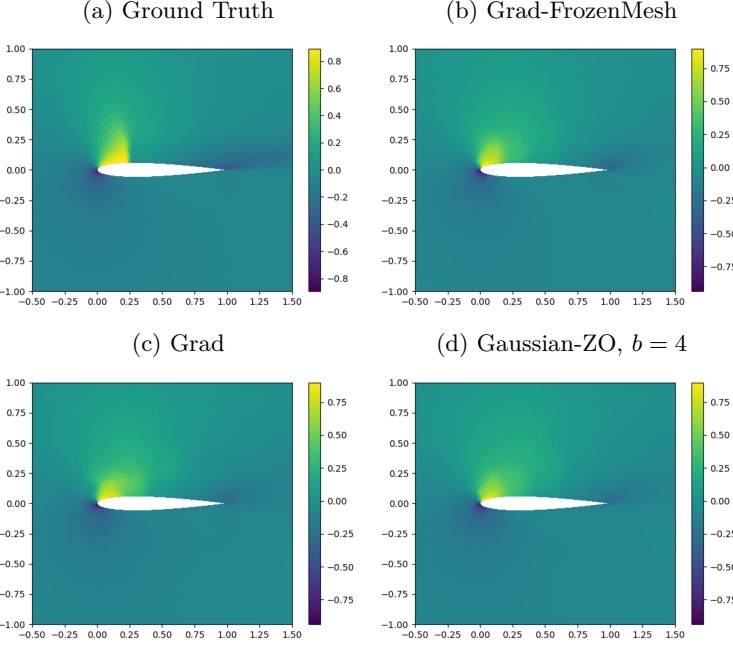

Figure 23: Predicted pressure fields for NACA0012 airfoil under AoA = −9.0 and mach number = 0.6.

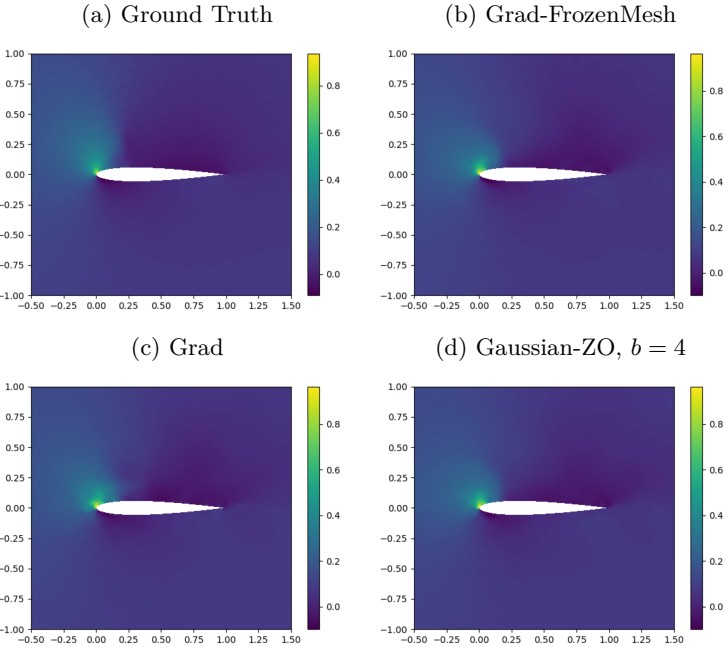

Figure 24: Predicted X-velocity fields for NACA0012 airfoil under AoA = −9.0 and mach number = 0.6.

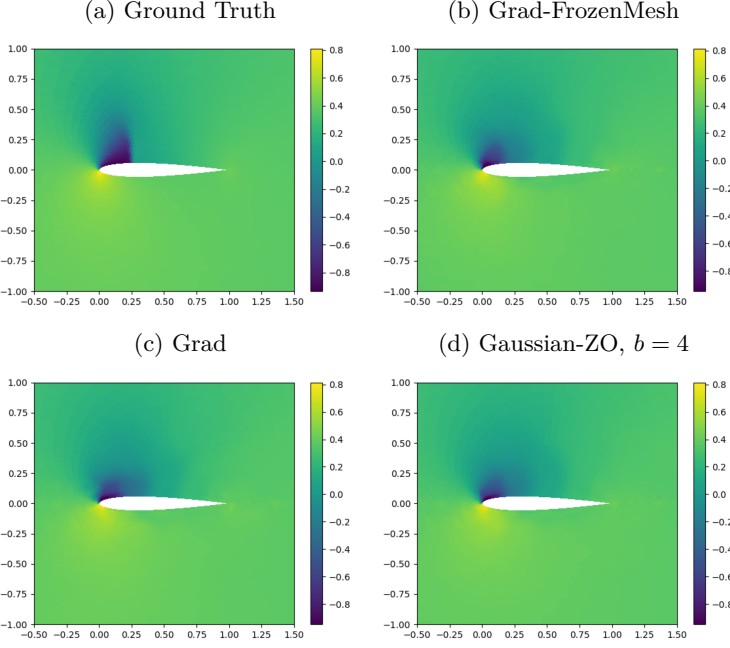

Figure 25: Predicted Y-velocity fields for NACA0012 airfoil under AoA = −9.0 and mach number = 0.6.

## C.3 Gaussian-Coordinate-ZO

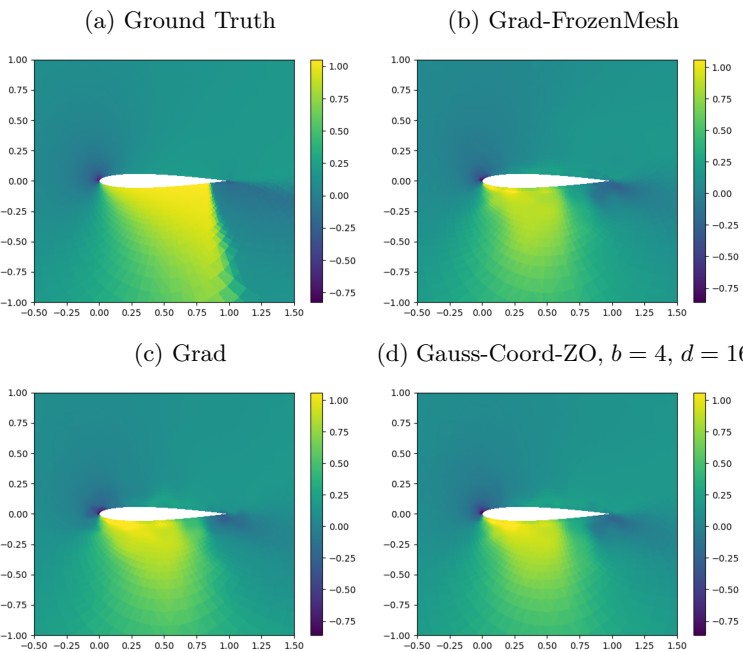

Figure 26: Predicted pressure fields for NACA0012 airfoil under AoA $= -10.0$ and mach number $= 0.8$.

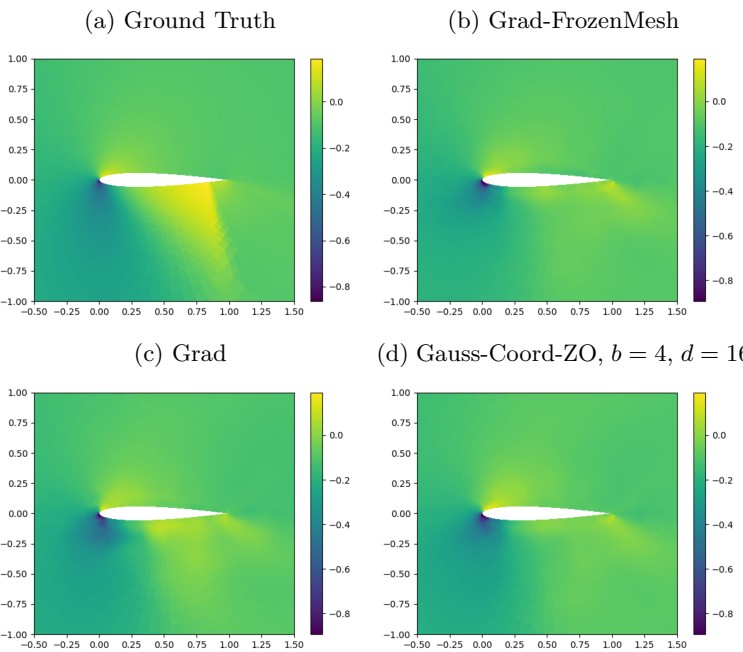

Figure 27: Predicted X-velocity fields for NACA0012 airfoil under AoA $= -10.0$ and mach number $= 0.8$.

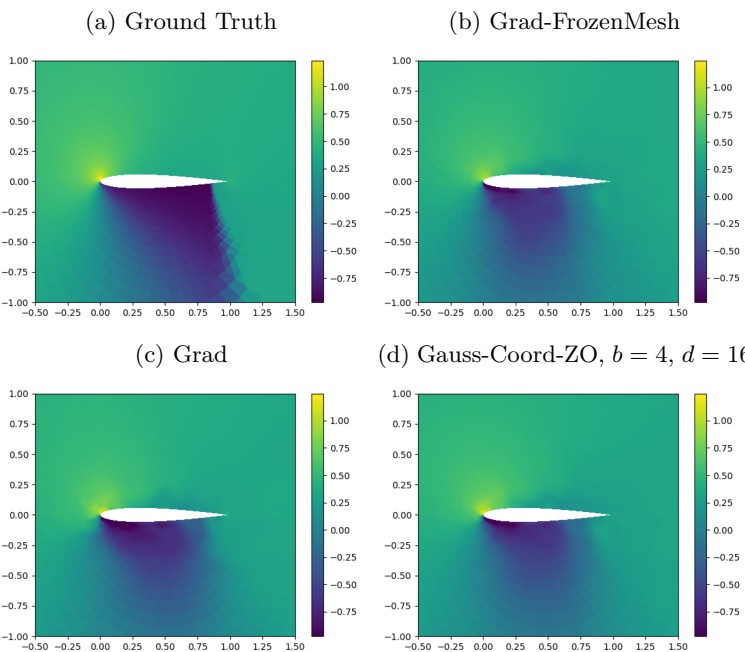

Figure 28: Predicted Y-velocity fields for NACA0012 airfoil under AoA = −10.0 and mach number = 0.8.

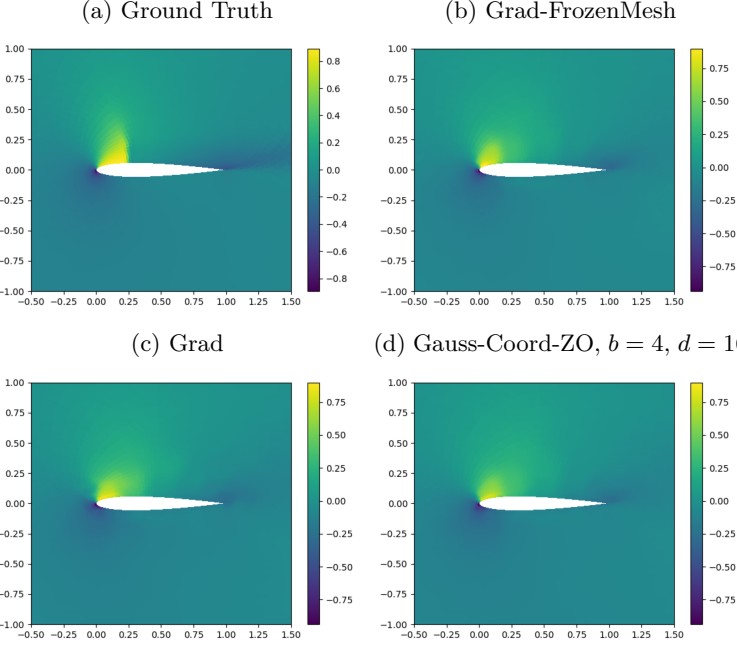

Figure 29: Predicted pressure fields for NACA0012 airfoil under AoA = −9.0 and mach number = 0.6.

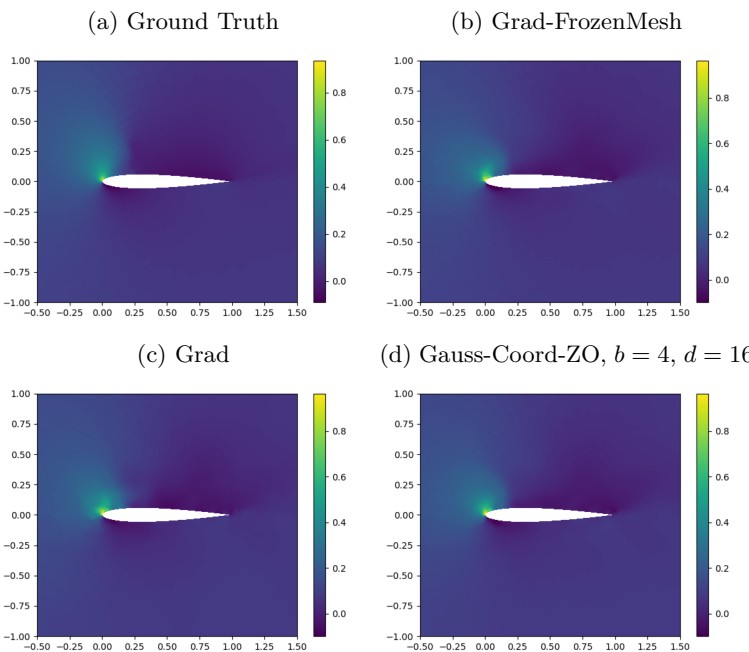

Figure 30: Predicted X-velocity fields for NACA0012 airfoil under AoA $= -9.0$ and mach number $= 0.6$.

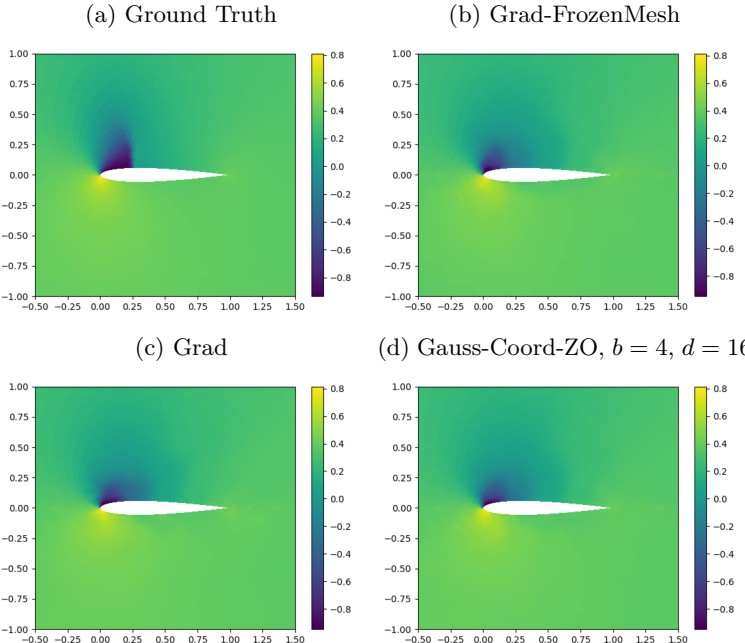

Figure 31: Predicted Y-velocity fields for NACA0012 airfoil under AoA $= -9.0$ and mach number $= 0.6$.

## D    Detailed Experimental Setting

In the experiments presented in this paper, we used a computing node equipped with two 18-core Intel Xeon E5-2695 v4 processors, 256 GiB of memory, and 2 NVIDIA P100 (Pascal) GPUs. During the training procedure, only one of equipped GPUs was used.

On the software front, our codes ran on the Tri-Lab Operating System Stack 4 (TOSS4) and employed Python 3.10.8, integrating key libraries like Pytorch (1.13.1+cu117) and SU2 (6.2.0). Before updating to TOSS4, we used older version of Python (3.9.12) and Pytorch (10.2.89) on TOSS3. For a fair comparison of the generalization performance of the hybrid system with different algorithms, we adhered to the same training and testing data split as the original implementation Belbute-Peres et al. (2020) and evaluated performance using the RMSE metric, consistent with CFD-GCN. Additionally, experiments involving different algorithms or parameters were performed once, using a consistent random seed to ensure reproducibility.

For repeating the experiment reported in this paper, please refer to the attached supplementary material ('README.md') for downloading the dataset, setting up the environment, training with the first-order or zeroth-order optimizer.

# E  Additional Experiments

In this section, we replace the SU2 solver with the MFEM solver (Anderson et al., 2021). This experiment highlights the capability of our proposed ZOO framework, in overcoming the lack of auto-differentiation for mesh coordinates in the PDE solver MFEM. We consider the following Poisson equation,

$$-\Delta(\alpha u) = 1.$$

The optimization problem we propose is formulated as follows:

$$\min_{\theta, M_{\text{coarse}}} \frac{1}{n} \sum_{i=1}^{n} \mathcal{L}\Big(\text{NN}_\theta(M_{\text{fine}}, O^i_{\text{coarse}}), O^i_{\text{fine}}\Big), \tag{8}$$

where $\theta$ is the parameter of the correction neural network used to correct the coarse mesh prediction. The train set contains 6 values of $\alpha$ in $\{0.9, 0.91, 0.92, 0.93, 0.94, 0.95\}$ and the test set contains 3 values of $\alpha$ in $\{0.905, 0.925, 0.945\}$.

## E.1  Comparison over Different Scales

In this subsection, we validate the performance of our proposed ZOO framework in varying scales. We specify the dimensions of the coarse mesh as $\{10 \times 10, 15 \times 15, 20 \times 20\}$. Correspondingly, the dimensions for the fine mesh are $\{100 \times 100, 150 \times 150, 200 \times 200\}$. The outcomes of this experiment demonstrate that our optimization framework, which substitutes the unknown gradient of external parameters with a zeroth-order approximation, achieves consistent and robust convergence across varying scales. As illustrated in Figure 32, the loss curve stabilizes after approximately 20 iterations for all considered scales. Employing a denser coarse mesh, such as $20 \times 20$, always leads to a closer approximation of the ground truth; so, it consistently achieves lower loss. This improvement is due to the fact that a denser coarse mesh captures more details compared to a less dense mesh.

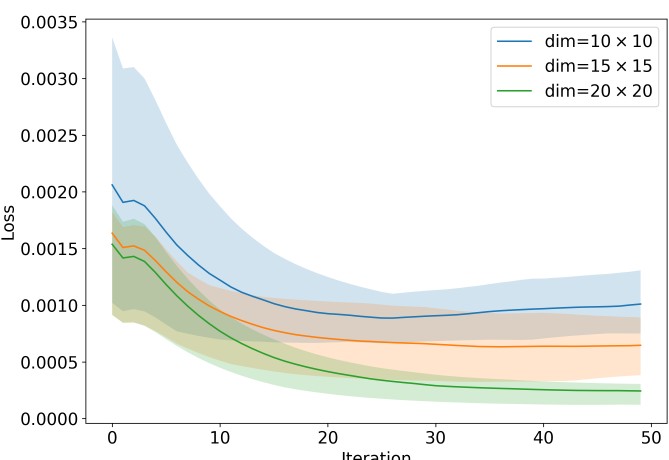

Figure 32: The comparison of loss curves among different scales. The loss curve represents the average of 10 training losses with 25% and 75% percentiles. Replacing the unknown gradient with the zeroth-order gradient estimation always leads to a decreasing loss curve for all considered scales.

## E.2   Results on Dynamic Flow

In this section, we assume the input $\alpha$ is dynamically changing. At each discrete time step $t$, $\alpha$'s value rises incrementally from 0.5 to 0.9 in the step-size of 0.1, which creates a time series $\alpha_1, \alpha_2, \alpha_3, \alpha_4, \alpha_5$. We solve the mesh optimization problem for each discrete time-step separately.

The Figure 33 illustrates the solution dynamic over the time $n$ before and after applying the gradient descent on both the mesh and the neural network for 50 iterations. We replace the gradient of the mesh with the zeroth-order estimation, the Gaussian-Coordinate-ZO method with the batch size $b = 4$ and the perturbation dimension $d = 16$. This result demonstrates the capability of our suggested optimization framework to adapt within dynamic settings, provided that the implementation structure is feasible in solving such problems.

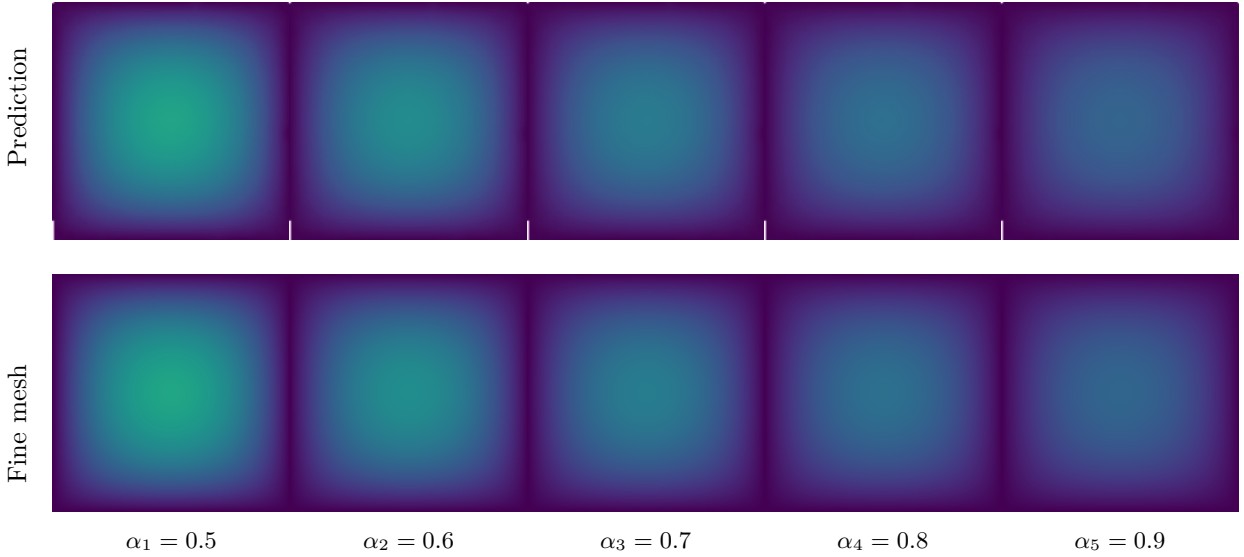

Figure 33: Comparison between the dynamic solution prediction over a fixed mesh.