# OpenReview forum: "End-to-End Mesh Optimization of a Hybrid Deep Learning Black-Box PDE Solver"
_TMLR — Rejected by TMLR_

### Review · Reviewer_Zsjk · 2024-02-04

**Summary Of Contributions:**

This paper focuses on the enhancement of hybrid deep-learning PDE solvers. The authors propose three models and one training strategy to tackle the gradient backpropagation issues for coarse mesh solvers, which can be used to refine the coarse mesh. Concretely, all the proposed methods are based on the classical zero-order gradient estimator. Experimentally, the proposed method can consistently outperform the frozen-mesh design and performs good generalization to the gradient-based method.

**Audience:**

Yes

**Broader Impact Concerns:**

This paper focuses on the technical problem, which can be inspiring to the design of hybrid PDE solvers.

**Claims And Evidence:**

Yes

**Requested Changes:**

The authors should add more implementation details and give new experiments about efficiency and new numerical solvers.

**Strengths And Weaknesses:**

## Strengths

1.	This paper focuses on an important and interesting problem.

2.	The proposed methods are reasonable with both theoretical analysis and experimental support.

3.	Overall, this paper is well-written. Both merits and weaknesses have been well discussed.

## Weaknesses

1.	About the presentation.

I am slightly confused about the training loss figures (Figures 2 and 5), especially the right subfigure. Why is the red curve stopped early? And in Figure 2, it is hard to tell which curve is the best. I think the orange curve (b=4) converges slower. Are there any explanations for this result?

2.	The proposed methods are too simple.

I think the zero-order method is quite intuitive and obvious. Two weaknesses of them are the training stability and the model efficiency. More discussions about these two items are expected, especially the model efficiency, which is highly related to the model practicability.

3.	Can the proposed methods apply to the other numerical solvers in addition to SU2?

Since the key idea of this paper is that the numerical solver could be hard to conduct automatic differentiation, it will be much better, if they also provide new experiments about another numerical solver.

---

> ### Author Response · Authors · 2024-03-09
>
> We sincerely appreciate the reviewer's feedback and valuable suggestions. We have thoroughly revised our paper with highlighting modifications in red based on these inputs.
>
> ***Q1***: About the presentation: I am slightly confused about the training loss figures (Figures 2 and 5), especially the right subfigure. Why is the red curve stopped early? And in Figure 2, it is hard to tell which curve is the best. I think the orange curve (b=4) converges slower. Are there any explanations for this result?
>
> ***A1***: Thank you for your detailed observations and questions regarding the training loss figures presented in our manuscript. We appreciate the opportunity to clarify these points.
>
> In Figures 2 and 5, the subfigures are designed to showcase different aspects of our analysis. The left subfigure in each figure illustrates the "Test Loss vs Number of Epochs," while the right subfigure focuses on the "Test Loss vs Number of Simulations." A larger batch size $b$ used in zeroth-order gradient estimation generally contributes to a more accurate approximation. Consequently, when we evaluate performance based on the number of epochs, a larger $b$ seems advantageous.  However, we note that this accuracy comes from running $b$ additional SU2 simulations. The practical implications of increasing $b$ become evident in the right subfigure, which plots "Test Loss vs Number of Simulations." This difference in scale is why the plots appear differently across the two subfigures.
>
> Specifically addressing your question about the red curve stopping early and the performance of the orange curve (b=4), we implemented a cutoff in the right subplots to emphasize efficiency in terms of simulations. Although the orange curve demonstrates superior performance over the same number of epochs (as shown in the left subplots), its higher simulation demand renders it less feasible for practical applications. We opted to truncate this curve where it reaches double the number of simulations compared to the first-order method. This cutoff was chosen as a practical balance, highlighting our focus on simulation efficiency.
>
> We hope this explanation resolves the confusion and better articulates the rationale behind our presentation choices in these figures.
>
> ***Q2***: The proposed methods are too simple: I think the zero-order method is quite intuitive and obvious. Two weaknesses of them are the training stability and the model efficiency. More discussions about these two items are expected, especially the model efficiency, which is highly related to the model practicability.
>
> ***A2***: Thank you for bringing up these critical aspects of zeroth-order methods. We agree that the stability and efficiency of these methods are paramount when considering their practical application and they are also the main challenges we are solving.
>
>  In many existing literature, it usually directly applies the zeroth-order method to the objective function [nesterov2017]. The zeroth-order method has an unimprovable lower bound with dependency on the input dimension $N$ (in our proposed framework, $N=352\times 2=704$). We propose using the zeroth-order gradient estimation to replace with the gradient of the single SU2 module instead of the whole CFD-GCN model; in this setting, the number of optimized parameters is significantly reduced compared to directly optimize the whole structure. By focusing the zeroth-order estimation on a narrower scope, we achieve more accurate gradient approximations than applying the method to the entire model's gradient.
>
>  [nesterov2017] Nesterov, Yurii, and Vladimir Spokoiny. "Random gradient-free minimization of convex functions." Foundations of Computational Mathematics 17 (2017): 527-566.
>
> ***Q3***: Can the proposed methods apply to the other numerical solvers in addition to SU2?
>
> ***A3***: Thank you for your inquiry regarding the applicability of our method to non-differentiable solvers. To address this question, we have conducted an additional experiment focusing on the Poisson equation:
> $$\Delta (\alpha u) = 1.$$
> For this purpose, we utilized PyMFEM [mfem2020], a solver without auto-differentiation capabilities for optimizing mesh coordinates at the current time point. By replacing the gradient with respect to the mesh with our Gaussian-Coordinate-ZO gradient estimation, we observe a stable and consistent decrease in the loss curve until convergence.  These findings demonstrate the adaptability of our method when applied to other solvers, particularly those that do not inherently support auto-differentiation. This experiment and its results have been added to the appendix of our revised paper.
>
> [mfem2020] Anderson, Robert, Julian Andrej, Andrew Barker, Jamie Bramwell, Jean-Sylvain Camier, Jakub Cerveny, Veselin Dobrev et al. "MFEM: A modular finite element methods library." Computers \& Mathematics with Applications 81 (2021): 42-74.

---

### Review · Reviewer_Ky2x · 2024-02-07

**Summary Of Contributions:**

For many years, solvers have been used to compute the solution to instances of PDE. Solvers are often slow and have undesirable scaling-up characteristics to large discretization, which are needed to compute accurate solutions. A line of work suggests generating data using solvers and using AI to learn the solution operator.

Another line of work suggests using a solver at low resolutions and training a deep-learning model as a corrector.

This current work is on the latter area.
------
Often, the solvers are given the input function at a discretization (mesh), and they solve the problem using that mesh.

Prior works suggest not only training the corrector but also learning the mesh through an optimization step. For such optimization that takes place using autograd, the solvers need to be differentiable. However, many solvers, especially those used in practice, are not written such that they are differentiable. This work suggests computing the gradient using sampling methods.

**Audience:**

Yes

**Broader Impact Concerns:**

This work can help advancing the field of AI for scientific computing.

**Claims And Evidence:**

No

**Requested Changes:**

Please kindly develop experimental study settings reflecting the limitations of state-of-the-art operator learning in scientific computing.

**Strengths And Weaknesses:**

Strenght,
The paper is based on prior ideas, and zeroth-order methods are valid. The presentation is good.

Weakness.
The main contribution of the work is showing how the method works with a black box solver in the loop. However, the empirical study can be made more convincing. Let me elaborate.

1) Scale. The zeroth order methods are computationally challenging when we work in real-world settings where the grid sizes are large. This needs to be addressed in the empirical study.

2) The work learns only one mesh. However, solvers can design a mesh per instance with their adaptive meshers. Please adapt your empirical study to the setting where the mesh is learned per instance to reflect challenges in practice (e.g., keep the oncoming air flow direction on the x-axis and move the airfoil or study change in the airfoil size).

3) The approach is only shown to work for the setting for which the range of coarse resolutions is similar, and the range of fine resolutions is similar. How does this approach work if, in the inference time, one can afford to provide the solver solution at a much higher resolution or slightly out of range?? In PDEs, we often deal with functions, and approaches are often needed to be mesh agnostic.

4) How does this approach address many temporal PDEs? e.g., Navier Stocks in time rather than time-averaged solution? In such cases, the input function is provided on a given mesh, and the solver gives the solution on the same mesh. Moreover, how the approach would work if, at the test time, we deal with different resolutions?

5) Following the above point, how does the method address the problem of many maps? For a given low-resolution solution, there are often infinitely many possible high-resolution solutions. How do we address that?

5) In many PDEs with coefficient functions or boundary functions as inputs, the input functions are provided only on the given mesh. The function is not provided on any other points. How do we handle those? In the current experiments, the input seems to be a few scalers describing the intel velocity direction and the velocity. Please consider advancing the setting of the empirical study to consider function as an input, e.g., temporal Navier Stokes or Darcy flow.

6) I encourage the authors to cover PDE literature, specifically those on operator learning and learning maps between function spaces, which is a predominant setting in PDEs.

Neural Operator: Learning Maps Between Function Spaces With Applications to PDEs
Fourier Neural Operator for Parametric Partial Differential Equations
Transformer for Partial Differential Equations' Operator Learning
GNOT: A General Neural Operator Transformer for Operator Learning


These works show sample efficient methods that have already been applied to highly complex PDEs, including automotive aerodynamics, weather forecasts, and many more. I am not asking the authors to run experiments in these settings; I am asking the authors to kindly develop experimental study settings reflecting the limitations of state-of-the-art operator learning in scientific computing.

---

> ### Author Response · Authors · 2024-03-09
> **Responses to Reviewer Ky2x (Part I)**
>
> We sincerely appreciate the reviewer's feedback and valuable suggestions. We have thoroughly revised our paper with highlighting modifications in red based on these inputs.
>
> ***Q1***: Scale. The zeroth order methods are computationally challenging when we work in real-world settings where the grid sizes are large. This needs to be addressed in the empirical study.
>
> ***A1***: Thank you for your insightful suggestion regarding the scalability of zeroth-order methods. To address this concern, we have included an additional experiment in the appendix, showcasing the applicability of our method across different scales. In this experiment, we focus on solving the Poisson equation:
> $$\Delta (\alpha u) = 1.$$
> The scale of the coarse mesh varies from $10\times 10$, $15\times 15$, and $20\times 20$; the scale of the corresponding fine meshes varies from $100\times 100$, $150\times 150$, and $200\times 200$, respectively. By replacing the gradient with respect to the mesh with our Gaussian-Coordinate-ZO gradient estimation, we observe a stable and consistent decrease in the loss curve until convergence. This empirical evidence supports the scalability of our proposed zeroth-order optimization framework.
>
> ***Q2***: The work learns only one mesh. However, solvers can design a mesh per instance with their adaptive meshers. Please adapt your empirical study to the setting where the mesh is learned per instance to reflect challenges in practice.
>
> ***A2***: Thank you for your insightful suggestion. To address this, we have added an additional experiment in the appendix, tailored to a dynamic setting. In this new experiment, we consider the Poisson equation:
> $$\Delta (\alpha_n u) = 1.$$
> where $\alpha_n$ represents the dynamic changing physical parameters at discrete time steps $n=1,2,3,4,5$. For each of these time steps, the model predicts the solution using a fixed mesh for different $\alpha_n$. This approach demonstrates the flexibility and capability of our proposed method to not only handle static scenarios but also dynamically adapt to changing conditions.
>
> ***Q3***: The approach is only shown to work for the setting for which the range of coarse resolutions is similar, and the range of fine resolutions is similar. How does this approach work if, in the inference time, one can afford to provide the solver solution at a much higher resolution or slightly out of range?? In PDEs, we often deal with functions, and approaches are often needed to be mesh agnostic.
>
> ***A3***:  Thank you for your insightful query regarding the adaptability of our approach to scenarios where the resolution of the solver's solution can vary significantly, especially beyond the initially considered range. The main challenge, as you've identified, relates to the fixed architecture of the GCN within the CFD-GCN model, which indeed necessitates a predefined fine mesh for input. Therefore, our current CFD-GCN-ZO model cannot handle such out-of-range scenarios.
>
> Our zeroth-order optimization framework, however, is designed with a broader applicability. Specifically,  if the underlying model is mesh agnostic while requires to update the external parameters which is not auto-differentiable, we can always replace the inaccessible  gradients for external modules with our zeroth-order gradient estimation.
>
> ***Q4***: How does this approach address many temporal PDEs? e.g., Navier Stocks in time rather than time-averaged solution? In such cases, the input function is provided on a given mesh, and the solver gives the solution on the same mesh. Moreover, how the approach would work if, at the test time, we deal with different resolutions?
>
> ***A4***: These scenarios will be addressed with replacing the backbone architecture CFD-GCN with a more flexible underlying model.  Still, when updating non-auto-differentiable external parameters is required in this hybrid structure, our methods are also applicable.
>
>
> ***Q5***: Following the above point, how does the method address the problem of many maps? For a given low-resolution solution, there are often infinitely many possible high-resolution solutions. How do we address that?
>
> ***A5***: We agree that for a given low-resolution solution, there can be infinitely many high-resolution solutions. In our CFD-GCN-ZO model, we leverage the up-sampling layer from the original CFD-GCN model, which employs a specific interpolation algorithm for generating high-resolution outcomes from low-resolution inputs.  Since this layer doesn't include any additional trainable parameters, it is not the primary focus of our work.

---

> > ### Author Response · Authors · 2024-03-09
> > **Responses to Reviewer Ky2x (Part II)**
> >
> > ***Q6***: In many PDEs with coefficient functions or boundary functions as inputs, the input functions are provided only on the given mesh. The function is not provided on any other points. How do we handle those? In the current experiments, the input seems to be a few scalers describing the intel velocity direction and the velocity. Please consider advancing the setting of the empirical study to consider function as an input, e.g., temporal Navier Stokes or Darcy flow.
> >
> > ***A6***:  Thank you for your insightful feedback. The CFD-GCN model's design, which relies on GCN components, requires inputs to include constant physical parameters (AoA and Mach number) at each mesh node. This design inherently limits the model's ability to directly process functional inputs.
> >
> > However, we believe the underlying methodology of our zeroth-order optimization approach (i.e. utilizing zeroth-order gradient estimation to replace the gradient of non-auto-differentiable parameters) remains applicable and valid even in a functional input setting.
> >
> > ***Q7***: I encourage the authors to cover PDE literature, specifically those on operator learning and learning maps between function spaces, which is a predominant setting in PDEs.
> >
> > ***A7***: We appreciate your suggestion to cover PDE literature on operator learning and learning maps between function spaces. We have added an overview of these results in the Related Work section.

---

### Review · Reviewer_pKd3 · 2024-02-23

**Summary Of Contributions:**

The authors continue the work of CFD-GCN where a differentiable PDE solver solves the problem on a coarse grid and a GNN uses the coarse solution to obtain a fine solution. The authors rightfully claim that most solvers are not differentiable, hence they wish to approximate the gradient of the PDE solver, including the coarse mesh, using zero-order sampling methods.

**Audience:**

Yes

**Broader Impact Concerns:**

No concerns.

**Claims And Evidence:**

Yes

**Requested Changes:**

Writing: it would be nice to show the reader the actual equation that you wish to solve, including boundary conditions.

All figures: it seems that as you increase b or d, the gradient approximation improves and so is the accuracy. What is the b that is needed to obtain the same accuracy as the method with the gradient? Can the authors demonstrate what happens as you further increase the samples?

Figure 3,6,9 etc: the error is not clear this way. Please also show the absolute error, so the difference between the approximations is clear.

Figure 4: If I understand correctly, these are the coarse mesh points. Can the authors draw the coarse meshes themselves? It doesn’t look good this way.

**Strengths And Weaknesses:**

*Strengths*:

The authors touch on a crucial and practical point in the CFD-GCN approach.

*Weaknesses*:

1) Figure 3,6 demonstrate that the errors are very high. Can the authors correct that? Maybe by using a network with higher capacity? What is the point in using NN’s to obtain such inaccurate solutions?

2) The authors showed their results using a solver which is in principle differentiable in SU2. What happens when the solver is not differentiable? Can the authors demonstrate their work with another off-the-shelf solver?

3) If there is a clear advantage to one of the gradient approximation methods, the authors do not emphasize it well.
The description about the parameters that are considered for the SU2 solver are not clear. What are the parameters of the solvers beyond the coarse mesh?

4) Comparisons: The authors cite quite a few works on deep learning methods for CFD, but compare only to CFD-GCN which is the baseline for their work. What about other works?

5) Missing references on data driven PDE solvers and especially hybrid methods:
-	Learning nonlinear operators via DeepONet based on the universal approximation theorem of operators.
-	A Hybrid Iterative Numerical Transferable Solver (HINTS) for PDEs Based on Deep Operator Network and Relaxation Methods
-	CFDNet: a deep learning based accelerator for fluid simulations
-	Multigrid-augmented deep learning preconditioners for the Helmholtz equation

---

> ### Author Response · Authors · 2024-03-09
> **Responses to Reviewer pKd3 (Part I)**
>
> We sincerely appreciate the reviewer's feedback and valuable suggestions. We have thoroughly revised our paper with highlighting modifications in red based on these inputs.
>
> ***Q1***: Figure 3,6 demonstrate that the errors are very high. Can the authors correct that? Maybe by using a network with higher capacity? What is the point in using NN’s to obtain such inaccurate solutions?
>
> ***A1***: We agree that the error shown in these figures is far from the ground truth (the solution over the fine mesh), and this result is actually as expected. We note that even using the first-order method (i.e., the original baseline CFD-GCN model), we cannot further reduce such error due to the following two reasons: (1) The GCN component cannot handle out-of-distribution data accurately. According to [Belbute-Peres2020] (Section 4.1 and Section 4.2), the neural network is better suited to predict the data that has been seen in the train set; however, in these two selected figures, we are using data from the out-of-distribution sample. (2) The coarse mesh is too coarse to provide enough correction information. Our selected coarse mesh only has 354 nodes, only 5\% of the fine mesh, making it hard to correct information within the grid where no additional physical information is provided. These two reasons limit the best possible performance gain from further optimizing the model.
>
> We still choose to include these figures because our primary focus is not to improve the accuracy of the CFD-GCN model. Instead, our main focus is to design a new zeroth-order optimization framework such that when the external PDE solver doesn't support the auto-differentiation, we can still achieve a similar performance as it supports. We believe the current Figure 3 and Figure 6 can clearly reflect it.
>
> [Belbute-Peres2020] Belbute-Peres, Filipe De Avila, Thomas Economon, and Zico Kolter. "Combining differentiable PDE solvers and graph neural networks for fluid flow prediction." In international conference on machine learning, pp. 2402-2411. PMLR, 2020.
>
> ***Q2***: What happens when the solver is not differentiable? Can the authors demonstrate their work with another off-the-shelf solver?
>
> ***A2***:   Thank you for your inquiry regarding the applicability of our method to non-differentiable solvers. To address this, we have conducted an additional experiment focusing on the Poisson equation:
> $$\Delta (\alpha u) = 1.$$
> For this purpose, we utilized PyMFEM [mfem2020], a solver without auto-differentiation capabilities for mesh coordinates at the current time point. By replacing the gradient with respect to the mesh with our Gaussian-Coordinate-ZO gradient estimation, we observe a stable and consistent decrease in the loss curve until convergence.  These findings demonstrate the adaptability of our method when applied to other solvers, particularly those that do not inherently support auto-differentiation. This experiment and its results have been added to the appendix of our revised paper.
>
> [mfem2020] Anderson, Robert, Julian Andrej, Andrew Barker, Jamie Bramwell, Jean-Sylvain Camier, Jakub Cerveny, Veselin Dobrev et al. "MFEM: A modular finite element methods library." Computers \& Mathematics with Applications 81 (2021): 42-74.
>
> ***Q3***: If there is a clear advantage to one of the gradient approximation methods, the authors do not emphasize it well. The description about the parameters that are considered for the SU2 solver are not clear. What are the parameters of the solvers beyond the coarse mesh?
>
> ***A3***:  To answer what the parameters are, we have added the actual equation that we wish to solve with the given boundary conditions.  Our goal is to solve the Navier-Stokes PDEs:
> $$\frac{\partial V}{\partial t}+\nabla \cdot \bar{F}^c(V)-\nabla \cdot \bar{F}^v(V, \nabla V)-S=0,$$
> where AoA is used to define the boundary condition (the flow-tangency Euler wall boundary condition for airfoil and the standard characteristic-based boundary condition for the farfield) and Mach number is used to define the initial condition to describe the initial velocity. They are used as a scalar input to both the PDE solver and the GCN component.

---

> > ### Author Response · Authors · 2024-03-09
> > **Responses to Reviewer pKd3 (Part II)**
> >
> > ***Q4***: Comparisons: The authors cite quite a few works on deep learning methods for CFD, but compare only to CFD-GCN which is the baseline for their work. What about other works?
> >
> > ***A4***:  Thank you for your valuable observation. Our primary objective is to develop a framework for updating non-auto-differentiable parameters in external PDE solvers. Most of the cited works in the deep learning domain for CFD may require additional physical simulators but do not have the specific requirement to update parameters in these external solvers.
> >
> > Specifically, the CFD-GCN model, which we selected as our baseline, contains a mesh optimization procedure; when the external PDE solver operates as a black-box system, the mesh optimization problem naturally reduces to a black-box optimization problem. This particular issue aligns closely with the core problem our work aims to solve, making CFD-GCN an ideal underlying model for testing our zeroth-order optimization framework.
> >
> > ***Q5***: Missing references on data driven PDE solvers and especially hybrid methods:
> >
> > ***A5***: Thank you for highlighting the omission. In response to your suggestion, we have enriched the Related Work section by incorporating additional references that discuss data-driven PDE solvers, with a particular emphasis on hybrid methods.
> >
> > *Requested changes*:
> >
> > Thank you for your constructive suggestions. Here's how we have addressed each of your points in our revised submission. (1) Equation and Boundary Conditions: We have now included the specific equation we aim to solve, along with the boundary conditions, early in the manuscript for clarity. (3) Error Clarification in Figures 3, 6, 9: To address your concern about error visibility, we've revised these figures to include plots of the absolute error. (4) Coarse Mesh Representation in Figure 4: We've taken your feedback into consideration and revised Figure 4 to the coarse meshes themselves, rather than just the points.
> >
> > In response to point (2) regarding the necessary batch size $b$ to achieve accuracy comparable to the first-order method, we refer to the lower bound of zeroth-order optimization [nesterov2017]. This result suggests that to match the same accuracy as the first-order method, a zeroth-order approach necessitates the complexity that is $N$ times larger, where $N$ denotes the dimensionality of the input. Given our input dimension is $N=352 \times 2 = 704$, achieving equivalent accuracy would require a batch size on the order of $704$; that is, $b=\Theta(N)$. Such a batch size surpasses our computational capabilities. We have incorporated this analysis and its implications into the revised manuscript, providing a justification for the limitations encountered in our approach.
> >
> >
> > [nesterov2017] Nesterov, Yurii, and Vladimir Spokoiny. "Random gradient-free minimization of convex functions." Foundations of Computational Mathematics 17 (2017): 527-566.

---

### Decision · Action_Editor_dZKo · 2024-04-01

**Recommendation:** Reject

**Comment:**

The paper, focusing on a crucial aspect of Computational Fluid Dynamics (CFD) using the CFD-GCN approach, delves into enhancing generalization in fluid dynamics simulations. While the paper offers theoretical and empirical backing for its methods, reviewers have pointed out several areas needing major revisions. Concerns include the need for a deeper integration with and differentiation from existing literature, along with a call for more comprehensive and comparative empirical analysis (e.g., temporal PDEs) to support their claim.

**Audience:**

Yes, the AI for science community in TMLR's audience would be interested in knowing the findings of this paper.

**Claims And Evidence:**

Yes, most of the claims made in the submission are supported by evidence that is clearly presented. However, the reviewers contend that the work does not sufficiently engage with existing literature, suggesting that the claims of novelty are somewhat overstated and not adequately substantiated.

**Resubmission Of Major Revision:**

The authors may consider submitting a major revision at a later time.